# Pan-cancer single-cell analysis reveals the heterogeneity and plasticity of cancer-associated fibroblasts in the tumor microenvironment

Cancer-associated fibroblasts (CAFs) are the predominant components of the tumor microenvironment (TME) and influence cancer hallmarks, but without systematic investigation on their ubiquitous characteristics across different cancer types. Here, we perform pan-cancer analysis on 226 samples across 10 solid cancer types to profile the TME at single-cell resolution, illustrating the commonalities/plasticity of heterogenous CAFs. Activation trajectory of the major CAF types is divided into three states, exhibiting distinct interactions with other cell components, and relating to prognosis of immunotherapy. Moreover, minor CAF components represent the alternative origin from other TME components (e.g., endothelia and macrophages). Particularly, the ubiquitous presentation of endothelial-to-mesenchymal transition CAF, which may interact with proximal SPP1+ tumor-associated macrophages, is implicated in endothelial-to-mesenchymal transition and survival stratifications. Our study comprehensively profiles the shared characteristics and dynamics of CAFs, and highlight their heterogeneity and plasticity across different cancer types. Browser of integrated pan-cancer single-cell information is available at https://gist-fgl.github.io/sc-caf-atlas/.

Along with malignant cells, heterogeneous tumor microenvironment (TME) composites are important parts of tumors. They interact with malignant cells and contribute to the hallmarks of cancer[1,2]. Different types of nonmalignant cells are present in the TME, mainly including fibroblasts, immune cells (e.g., myeloid cells, lymphocytes, and macrophages), and endothelial cells. Previous studies have highlighted the indispensable role of the TME in the biological capabilities of cancer, such as tumor progression, treatment resistance, angiogenesis induction, and metastasis[2–4]. Mechanistically, the TME influences cancer cells via complicated and dynamic pathways to regulate cancer-related signaling[5], involving ligand–receptor interactions (e.g., PD-L1 of cancer cells binding to PD1 of T cells), cytokine/metabolite responses, and deposition of extracellular matrix (ECM)[6–10].

Of all types of stromal cells, fibroblasts are the predominant component in the TME, and cancer-associated fibroblasts (CAFs) play prominent and diverse tumor-supporting roles in cancer[10,11]. Besides directly interacting with malignant epithelial cells, CAFs create a tumor-permissive TME, including inducing the activation of normal fibroblasts into CAFs, promoting angiogenesis with endothelial cells[12], recruiting myeloid cells[13], and immunosuppression of T cells[14]. Therefore, CAFs play a key role in sculpting the TME by interacting with other TME components[7,10,11], exhibiting their potential value as a prognostic factor and therapeutic target[15]. On the other hand, CAFs are primarily derived from normal fibroblasts (NFs) via several cancer type-specific mechanisms, and the modulation of CAFs to the TME is diverse and vague due to their heterogeneity and plasticity[7]. Recent studies have specified various subtypes of CAFs[7], but the definitive

✉e-mail: luohan-hx@scu.edu.cn; jihwan.park@gist.ac.kr; xuheng81916@scu.edu.cn

origin of CAFs is controversial, and the lack of generalized characterization hinders CAF-targeted therapy in clinical practice. Meanwhile, it is difficult to define small subtypes of CAFs and investigate the plasticity of CAFs with limited cell populations, thus hindering the investigation on the evolutionary trajectory of CAFs and possible switches between CAFs and other TME components during cancer development and progression.

In recent years, the development of single-cell RNA sequencing (scRNA-seq) technology has provided the opportunity to investigate fluctuations in cell status and the strength of cell plasticity[15–17]. The characteristics of cancer cells and the TME have been profiled in multiple types of cancer, revealing the heterogeneity of cancer samples with different components at single-cell resolution and suggesting their possible involvement in the biological capabilities of cancer[11]. Additionally, increasing scRNA-seq-related software and strategies have been developed to improve the accuracy of bioinformatics analysis and upgrade the analytical dimensions, including batch effect correction[18], cell–cell interaction evaluation[19], and evolutionary trajectory[20]. However, recent scRNA-seq studies are still limited by the sample size, mainly due to fresh sample availability and expenditure. On the other hand, a large sample size would facilitate characterizing the complexity of cancer by excluding individual variants and by enriching the distinct components with a small cell population, particularly those shared across different cancer types. Due to the similarity of TME cells but not cancer cells among patients with different cancer types[21,22], a few scRNA-seq-based pan-cancer studies have recently been conducted to integrate the increasing accessibility of scRNA-seq profiles, thus maximizing resolution and analyzing cell quantity within controlled bias, and exhibiting the ubiquitous characteristics of TME cells[21–24]. However, these pan-cancer studies have only focused on the characteristics of immune cells, omitting the interactions between different cell components.

In this study, we combine public and inhouse scRNA-seq data to profile the TME across 10 common solid cancer types, characterizing the interconvertibility and interaction among different types of stromal cells, with a particular focus on the ubiquitous characteristics of CAFs among diverse cancer types and the plasticity of CAF subtypes. Our systematic investigation on CAFs and their subtypes across cancers at single-cell resolution highlights the possible heterogeneity and plasticity of CAFs in cancer biology.

## Results

### Landscape of the TME in pan-cancer illustrated using scRNA-seq analysis

To profile the TME landscape of solid cancers, we compiled a single-cell transcriptional atlas in 10 common solid cancer types (Fig. 1a), with 148 primary tumor, 53 adjacent, and 25 normal samples from 164 donors enrolled in 12 studies (Fig. 1b and Supplementary Data 1). Available independent cohorts of patients with the same cancer type (i.e., pancreatic, lung and prostate cancers) were included to establish internal validation (Fig. 1b). Additionally, scRNA-seq data of the counterpart normal tissues from public resources were included to match the selected cancer type (Fig. 1b)[25]. To exclude possible technology-induced bias, all scRNA-seq data were generated using the same platform (i.e., 10× Genomics) without specific sorting. After strict quality control and filtering, a total of 855,271 cells from 226 samples were included and integrated based on a batch effect correction algorithm (Supplementary Fig. S1a). Next, unsupervised clustering generated a total of 34 TME-related clusters (c1-c34, 569,759 cells), which were separated from cancerous/normal epithelial cells (Fig. 1c and d). These clusters were divided into five major cell components based on canonical markers of different cell types, including fibroblasts (c1-c8, *DCN*, *COL1A1*), lymphocytes (c9-c17, *CD3D*, *CD3E*), myeloid cells (c18-c24, *CD68*, *CD14*), endothelial cells (c25-c28, *VWF*, *PECAM1*), and plasma cells (c29-c34, *IGHG1*, *JCHAIN*) (Fig. 1c, e and

Supplementary Data 2). The remaining cells were clustered as epithelial cells (*EPCAM*, *KRT19*) (Fig. 1c). No obvious bias of the TME was observed regardless of malignant state and tissue type (Supplementary Fig. S1b and c), compared to the tissue type-specific distribution of epithelial cancer cells (Supplementary Fig. S1d). The normalized proportion of different clusters varied substantially among the different malignant states and cancer types (Fig. 1d and Supplementary Data 3). Although the composition of the five major components was relatively homogenously distributed among normal, adjacent, and tumor tissues, cells from cancer/adjacent tissues were predominantly enriched in specific clusters (e.g., c2 and c9), and vice versa (e.g., c3) (Fig. 1e). Additionally, some clusters, particularly c20 (*FABP4*⁺ macrophages), exhibited cancer type enrichment, which was consistent with an independent study on lung cancer[26] (Fig. 1f). Intriguingly, a significant stepwise decrease in the proportion of *FABP4*⁺ macrophages (c20) but not other macrophage clusters (i.e., *IL1B*⁺, SPP1⁺, and *APOE*⁺) was observed along with adjacent controls, early- and late-stage primary tumors, and brain metastasis of lung cancer[26] (all $p < 0.001$ to normal tissue) (Fig. 1g and Supplementary Fig. S1e), suggesting that *FABP4*⁺ macrophages may negatively correlate with lung cancer progression.

After deeply investigating the components of the TME to evaluate its heterogeneity, potentially functional subclusters were identified. For instance, heterogeneity of *APOE*⁺ tumor-associated macrophages (TAM, c19) were exhibited by the complementary distribution of *C1QC*⁺ and SPP1⁺ cells (Fig. 1h), which is consistent with a previous report on pan-cancer scRNA-seq analysis[21]. The SPP1⁺ TAMs showed distinct transcriptomic profiles compared to *C1QC*⁺ TAMs, which are involved in several metabolism-related pathways (Supplementary Fig. S1f). The proportion of SPP1⁺*C1QC*⁻/SPP1⁺*C1QC*⁺ TAMs but not SPP1⁻*C1QC*⁺/SPP1⁻*C1QC*⁻ TAMs significantly increased from normal to adjacent/tumor samples among different cancer types (Fig. 1i), suggesting the potential role of SPP1⁺ TAMs in tumorigenesis and tumor metabolism. On the other hand, tissue preferential distribution of endothelial cells promoted their classification into tumor endothelial cells (TECs) (i.e., c25) and normal endothelial cells (NECs) (i.e., c26 and c28) (Fig. 1e). The distinct transcriptional profiles of TECs and NECs were identified, including TEC significantly upregulated genes involved in the insulin response (e.g., *INSR* and *IGFBP*s), MAPK regulators (e.g., *SPRY1*), and immunoglobulins (e.g., *CD320* and *IGHG4*)[27,28] (Fig. 1j), suggesting the possible role of TECs in angiogenesis, tumor growth, and immune modulation. Moreover, pan-cancer analysis enhanced the possibility of delineating the potentially important but low-quantity subsets of immune cells in the TME. For instance, tumor/adjacent confined suppressive dendritic cells (i.e., DC subcluster 5) have been identified as a subset of dendritic cells (c21) with high expression of *CD274*, *LAMP3*, and *CCL22*, which may negatively regulate immune cells[29,30] (Supplementary Fig. S1g). Intriguingly, small portions of both DCs and plasmacytoid DCs (pDCs, c24) expressed T-cell-specific markers (e.g., *CD3D* and *CD3E*) (Supplementary Figs. S1g and 1h). As validation, confocal images of multiplexed immunofluorescence (mIF) with co-staining CD3, CD86 and CD11c in stromal compartments of three cancer types (i.e., anaplastic thyroid cancer, gastric cancer and colorectal cancer), revealed the consistent existence of CD3⁺ DCs (CD3⁺CD11c⁺CD86⁺) (Supplementary Fig. S1i), ranging from 3 to 13% of myeloid cells (CD11c⁺) (Supplementary Fig. S1j), which has also been reported through flow cytometry[31]. On the other hand, heterogeneity of tumor-infiltrating B lymphocytes was found through subclustering. For instance, a subset of the B-cell cluster (i.e., c13) expressed markers consistent with suppression of B-cell differentiation (*RGS13*⁺, subclusters 7–8) (Supplementary Fig. S1k). Future studies should be conducted to investigate the origin and potential biofunction of these B-cell subpopulations in tumors.

Moreover, we also estimate the ubiquitous characteristics of epithelia among different cancer types. First, a total of 23 clusters (E1-E23) were divided with unsupervised clustering (Supplementary

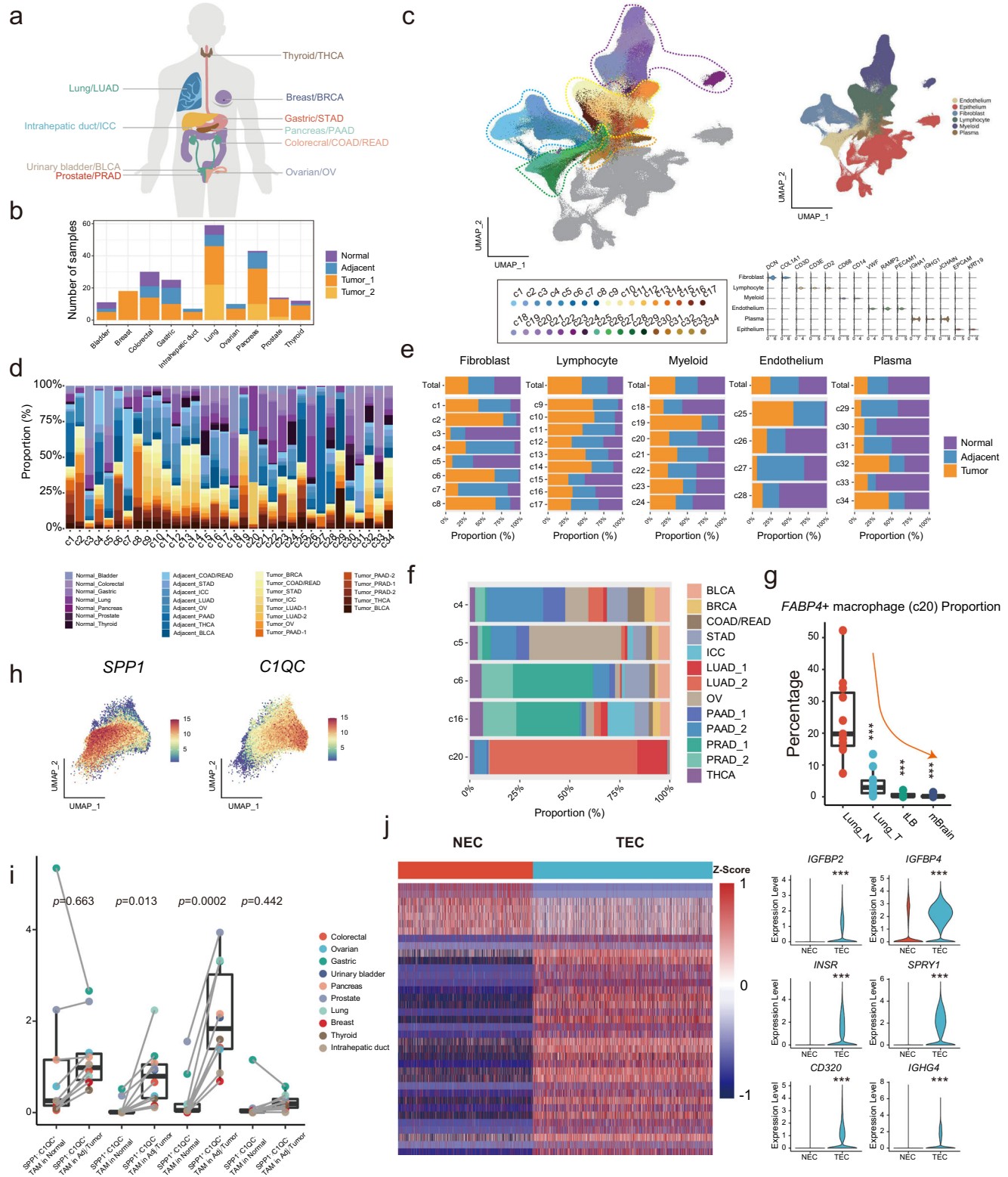

Fig. S2a and Supplementary Data 4). Unlike TME components, epithelia clusters exhibited bias in terms of both malignant status (e.g., pre-dominant normal in E17 and tumor in E6) and cancer type (e.g., pre-dominant thyroid cancer in E1 and prostate cancer in E6) (Supplementary Data 5 and Supplementary Fig. S2b). Not surprisingly, similarity among different cancer types was identified in only 6 out of 23 clusters, including E3 (*IRS2, KRT6A*), E5 (*CD24, STMN1*), E8 (*GKN1, MUC5AC*), E9 (*PHGR1, TFF3*), E10 (*TFF3, TPO*) and E13 (*VTN* and *ITIH5*) (Supplementary Fig. S2c, d). Interestingly, *IRS2* and *CD24* was widely

present in epithelia and exhibited the strongest expression in E3 and E5 cluster respectively (Supplementary Fig. S2d), suggesting the ubiqui-tous activated insulin signaling[32] and "don't eat me" signal[33] shared by different cancer types.

## Generalized activation of CAFs in the TME

Compared to the biased distribution of epithelial cells (Supplementary Fig. S1d), TME components from different cancer types were clustered together by unsupervised dendrograms. This proved that different

**Fig. 1 | Landscape of the TME in pan-cancer illustrated using scRNA-seq analysis. a** The cancer types included in this pan-cancer study. **b** The sample size histography of the selected normal/adjacent/tumor tissues. The sample size of replicates is shown when applicable. **c** Uniform Manifold Approximation and Projection (UMAP) plots of pan-cancer with 34 TME clusters, which are grouped into 4 main parts (i.e., endothelial cells, fibroblasts, lymphocytes/plasma cells, and myeloid cells). **d** Histography of the composition proportion of different tissue types in each TME cluster. **e** Clustering of TME components and their composition proportions in normal, adjacent, and tumor tissues. The proportion was normalized to the total cell number in each cancer. **f** Cancer type composition histography of tissue-enriched clusters according to cellular origin (c4-6, c16 and c20). **g** Significantly decreased proportions of $FABP4^+$ macrophages along adjacent normal lung (Lung_N, $n = 11$), lung tumor (Lung_T, $n = 11$), advanced stage of tumor (tLB, $n = 4$), and brain metastasized (mBrain, $n = 10$) tissues, The box is bounded by the first and third quartile with a horizontal line at the median and whiskers extend

to the maximum and minimum value. Dunnett-t two-sided test is used to test the significance of FABP4 + macrophages proportion between different tumor and normal tissue categories, Lung_N vs Lung_T $p$-value is $5.87 \times 10^{-7}$, Lung_N vs tLB, $p$-value is $1.26 \times 10^{-5}$, Lung_N vs mBrain, $p$-value is $2.97 \times 10^{-8}$; ***: $p < 0.001$. **h** Feature plot of SPP1 and C1QC expression in the tumor-associated macrophage cluster. **i** Comparison of $C1QC^+$ TAMs, $SPP1^+$ TAMs, $C1QC^+/SPP1^+$ TAMs and $C1QC^-/SPP1^-$ TAMs in normal and adjacent/tumor tissues, normal tissue $n = 43$, adjacent/tumor tissue $n = 159$. The box is bounded by the first and third quartile with a horizontal line at the median and whiskers extend to the maximum and minimum value. Mann–Whitney two-sided test is used to test the significance of proportion between different cell types. ***$p < 0.001$. **j** Differentially expressed genes clustering and specifically altered genes between tumor endothelial cells (TECs) and normal endothelial cells (NECs). Mann–Whitney two-sided test is used to test the significance of gene expression level between NEC and TEC categories. Source data are provided as a Source Data file.

lineages of TME components shared similar transcriptomic profiles across different cancer types (Supplementary Fig. S3a). Given complicated intercommunications among TME components play a critical role in tumor progression and treatment response, CellphoneDB-based analysis was conducted to evaluate the interactions between each major TME component and epithelial cells. The overall interactions significantly increased in the order of normal, adjacent, and tumor regardless of the tissue type ($p < 0.001$) (Fig. 2a). Notably, crosstalk among fibroblasts, endothelial cells, and myeloid cells was dominant in the TME, and fibroblasts presented the most prolific interactions with other TME components in tumor/adjacent samples regardless of the tissue type (all $p < 0.001$) (Fig. 2a, b and Supplementary Fig. S3b), suggesting the possible important role of fibroblasts in cancer biology through communication with other TME components.

A total of eight clusters (c1-c8) were annotated as fibroblasts (*ACTA2* and *ACTG2*) and presented in all tissue types (Fig. 1e). According to the differences in the distribution of cell proportions, cells in c3 and c5 were dominantly derived from normal tissues and referred to as NFs, while the remaining six tumor-enriched clusters were considered as CAFs due to their predominant deviation from tumor/adjacent tissues (Fig. 1e). Compared to the transcriptional profile of NFs, collagen activation- and matrix metalloproteinase-related genes were highly expressed in CAFs, suggesting the activation of fibroblasts in CAFs (Supplementary Fig. S3c and Supplementary Data 6). Furthermore, angiogenesis- and immunomodulation-related genes (e.g., *PDGFRA*, *PDGFRB*, *FAP*, *NOTCH3*, *HES4*, and *THY1*) were significantly upregulated in CAFs (Fig. 2c). Specifically, according to canonical markers[34–36] (Fig. 2d), the three quantity-predominant major components (i.e., c1, c2, and c4) were defined as cancer-associated myofibroblasts ($CAF_{myo}$), inflammatory CAFs ($CAF_{infla}$), and adipogenic CAFs ($CAF_{adi}$) by over-presenting *ACTA2*, *FAP/TGFB1*, and *CFD*, respectively (Fig. 2d and Supplementary Fig. S3d). Given the consensus in the origin of CAFs[7], the other three minor components (i.e., c6-c8) were identified as endothelial-to-mesenchymal transition CAF ($CAF_{EndMT}$), peripheral nerve-like CAF ($CAF_{PN}$), and antigen-presenting CAF ($CAF_{ap}$) by overpresenting specific marker genes (Fig. 2d and Supplementary Data 2). As expected, these CAF clusters are involved indistinct pathways (Supplementary Fig. S3e).

Furthermore, we determined the possible involvement of regulons in $CAF_{myo}$, $CAF_{infla}$, and $CAF_{adi}$ by SCENIC analysis. Tumorigenesis- (e.g., *TBX2*[37]) and myogenesis-related regulons (e.g., *MEF2C*[38]) were highly enriched in $CAF_{myo}$. In addition, dedifferentiation-related (e.g., *CREB3L1*[39]) and epithelial-mesenchymal transition (EMT)-related regulons (e.g., *TWIST2*[25]) were enriched in $CAF_{infla}$ and $CAF_{adi}$, respectively (Fig. 2e). Therefore, we speculated that the activation of $CAF_{myo}$ was different from $CAF_{adi}$ and $CAF_{infla}$. Through similarity analysis as described previously[21], the same major lineages of CAFs from different cancer types were clustered together, illustrating their shared characteristics among diverse cancer types (Supplementary Fig. S4a).

Therefore, we pooled the CAFs from all cancer types together to explore the possible general activation process of CAFs. Two distinct activation paths from NFs to CAFs were revealed via evolutionary trajectory, enhancing the definition of three different states, $CAF_{state1}$ (NFs dominant), $CAF_{state2}$ ($CAF_{myo}$ dominant), and $CAF_{state3}$ ($CAF_{adi}/CAF_{infla}$ dominant) (Fig. 2f), all of which were not biased in terms of the constitution of tissue type (Supplementary Fig. S4b). Both the EMT score (Supplementary Data 7) and *CREB3L1* expression gradually increased along the activation trajectory of CAFs and were significantly higher in $CAF_{state3}$ than in $CAF_{state1/2}$ ($p < 0.001$) (Fig. 2g and Supplementary Fig. S4c) regardless of the tissue type (Supplementary Fig. S4d), suggesting a general dedifferentiated process along CAF activation. Consistently, the regulon activity of *CREB3L1* (Supplementary Data 7) significantly increased along with the expression of *CREB3L1* and peaked in $CAF_{state3}$ ($p < 0.001$) (Supplementary Fig. S4c−e). Finally, tracing the gene fluctuation along biforked trajectories, $CAF_{state2}$ tended to act in angiogenesis represented by overexpressing *INS* and *PDGFRB*, while $CAF_{state3}$ had a high expression of both pro-angiogenic (e.g., *PDGFRA*)[40] and immunomodulation-related genes (e.g., *ENG* and *FAP*)[36] (Fig. 2h and Supplementary Data 8). Collectively, the major CAFs were derived from NFs and evolved into different differentiated states that may exhibit distinct effects on the TME.

## Interactions of CAFs with TME and epithelia

Immunosurveillance escape is one of the major hallmarks in cancer, and CAFs can facilitate this process not only by providing a physical barrier but also by impacting on the immune TME. Based on canonical gene expression, tumor-infiltrating- natural killer cell (NKC)/T-cells were divided into three NKC clusters (e.g., $TIM3^+$ NKCs) and eight T cell clusters (e.g., $CD8^+$ cytotoxic T cells and regulatory T cells (Tregs)) (Supplementary Fig. S5a). The interaction counts between $CAF_{state3}$ and NK/T subclusters were superior to $CAF_{state1}$ and $CAF_{state2}$ (Fig. 3a), highlighting its potential immunoregulatory role. Shared and specific reciprocal communication was observed between different CAF states and NK/T cells. In particular, *CD44*, an important index of T cell activation and navigation in antitumor immunity[41], exclusively interacts with *FGF2* in $CAF_{state1}$ among different NK/T subclusters, while TGF-β (*TGFB1*)[42], progranulin (*GRN*)[43], galectin-9 (*LGALS9*)[44], and tumor necrosis factor (*TNF*)[45], which regulate adaptive immune cells, tend to interact with their partner receptors in $CAF_{state2}$ and $CAF_{state3}$ (Fig. 3b). Subcluster-specific interactions were also observed, such as the absence of most interactions in double-negative T cells (DN T) and specific LTA_TNFRSF1A binding in Treg/cycling $CD4^+$ cells for all CAF states (Fig. 3b). In addition to interact with NK and T cells, CAFs may also regulate tumor-infiltrating B-cell by overexpressing the TNF ligand superfamily. For instance, $CAF_{state3}$ interacted with the majority of B-cell subclusters via TNFSF13B_TNFRSF13B/C (B-cell survival/maturation-related genes)[46] (Supplementary Figs. S1k and S5b). In addition to

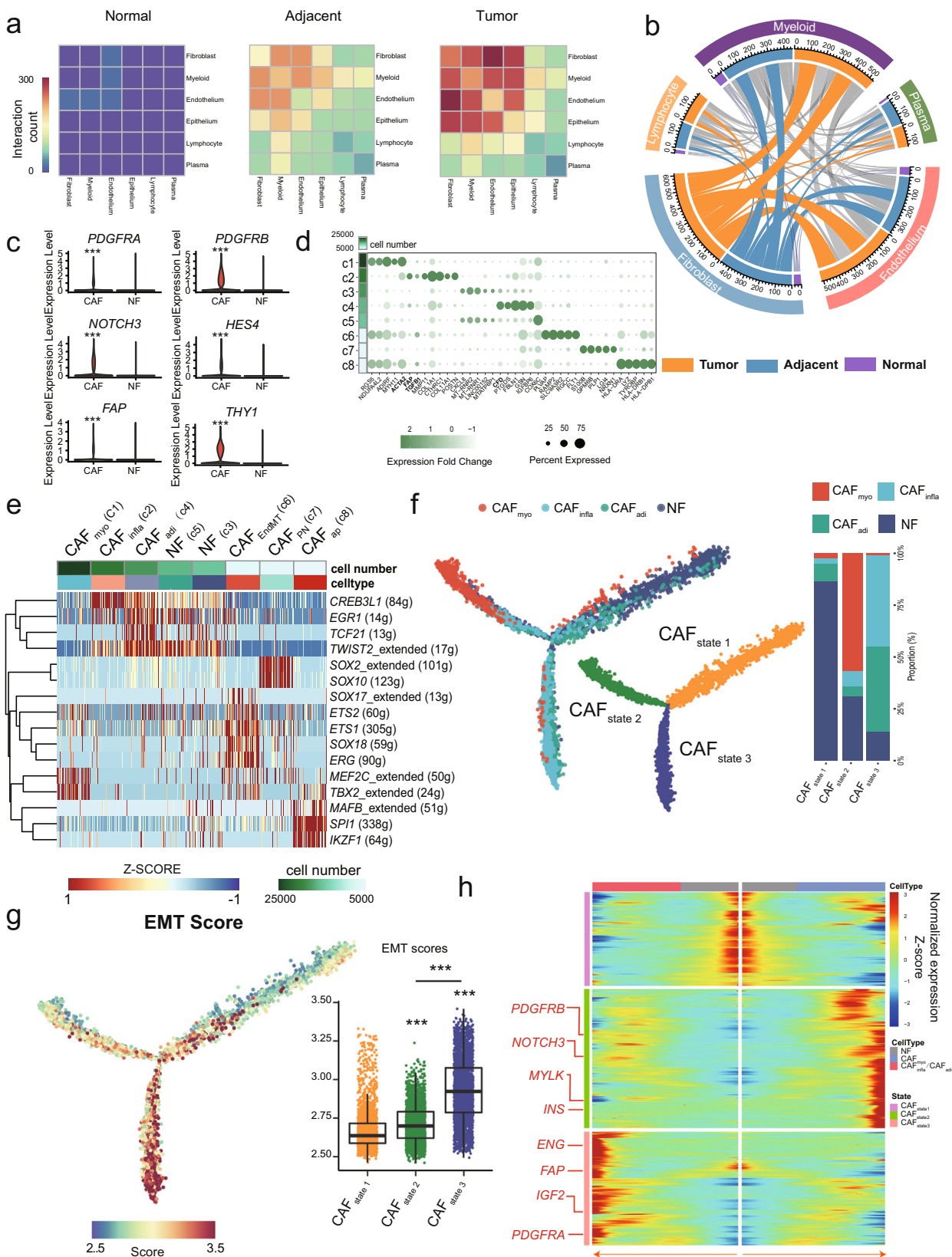

adaptive immunity, CAFs may also modulate innate immune cells, illustrating predominant interaction counts between CAF$_{state3}$ and myeloid components (Fig. 3c). TAM was the most prolific partner, whereas mastocytes hardly interacted with CAFs (Fig. 3d). Through the CXCL12_CXCR3 interaction, CAF$_{state1/2/3}$ specifically may recruit pDCs (Fig. 3d), which can secrete granzyme B to constrict the expansion of

T cells[47] (Supplementary Fig. S1h). In addition to the suppressive function of CAFs in priming DCs, CAF$_{state3}$ may also stimulate DC activation and maturation by interacting with galectin-9 (*LGALS9*)[48] and CD40 [ref. 49] on DC subtypes (Supplementary Fig. S5c). Therefore, CAFs, particularly CAF$_{state3}$, may present immunomodulatory capability.

**Fig. 2 | Generalized activation of CAFs in the TME. a** The mutual interaction among the main TME components and epithelial cells in different tissue origins. **b** The interaction between fibroblasts and other TME components. The length of arcs represents the predicted interaction counts. **c** Violin plot of specific marker genes in cancer-associated fibroblasts (CAFs) and normal fibroblasts (NFs). Mann−Whitney two-sided test is used to test the significance of gene expression level between CAF and NF categories. ***$p < 0.001$. **d** Bubble plot showing the expression of tag genes between CAFs and NFs. **e** Regulons enriched in each fibroblast cluster detected via SCENIC analysis. **f** Left: activation trajectory of CAFs,

which are divided into three states ($CAF_{State1/2/3}$). Right: histography of the different CAF components in each CAF state. **g** Left: epithelial-mesenchymal transition (EMT) score enriched along the evolutionary trajectory of CAFs. Right: comparison of EMT scores among three CAF states, state1: $n = 2130$, state2: $n = 4948$, state3: $n = 8667$. The box is bounded by the first and third quartile with a horizontal line at the median and whiskers extend to the maximum and minimum value. Mann−Whitney two-sided test is used to test the significance of EMT scores among different state categories. ***$p < 0.001$. **h** Fluctuation of genes along different states. Source data are provided as a Source Data file.

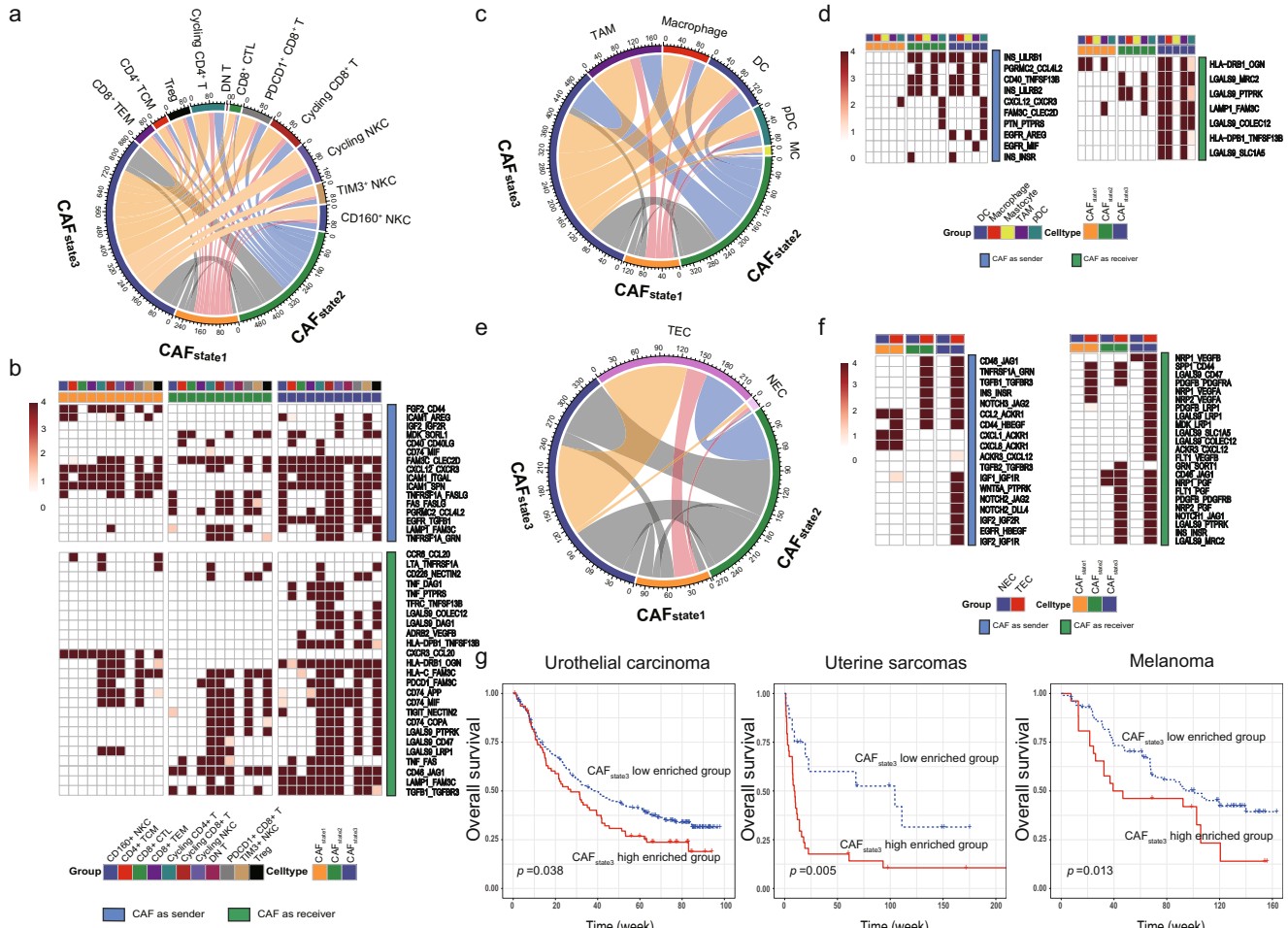

**Fig. 3 | CAFs orchestrate the immune TME and angiogenesis. a** and **b** Predicted and detailed interactions between different CAF states and NK/T cells. **c** and **d** Predicted and detailed interactions between different CAF states and subgroups of dendritic cells. **e** and **f** Predicted and detailed interactions between different CAF states and endothelial cells (TECs and NECs). **g** Estimation of the prognostic value

of the $CAF_{State3}$ signature score in three immunotherapy cohorts (urothelial carcinoma, uterine sarcoma, and melanoma) Kaplan−Meier curves for overall survival in all patients according to the number of positive ligands. $p$-values for all survival analyses have been calculated using the log-rank test.ue.

Next, we examined prime activity on endothelial cells, which are essential for angiogenesis. More dramatic interactions were observed between TECs and $CAF_{State2/3}$ than $CAF_{State1}$ (Fig. 3e), including NOTCH1/3_JAG1/JAG2, TGFB1_TGFBR3 and INS_INSR (Fig. 3f), which are involved in angiogenesis-related VEGF and NOTCH signaling pathways. In particular, interactions of NOTCH2_JAG2/DLL4, ACKR3_CXCL12 and IGF1/2_IGFR1R/2 R were specifically present between $CAF_{State3}$ and TECs.

Moreover, communications between three CAF states and each epithelia cluster were estimated. Interestingly, $CAF_{State3}$ also exhibited more interactions with epithelia than $CAF_{State1/2}$, especially with E3/E13 (clusters shared by all cancer types), E4 (dominant by digestive system tumor), and E18 (dominant by breast and ovarian cancer)

(Supplementary Figs. S2b and S5d). Moreover, we focused on the crosstalk between each CAF state with the epithelia clusters exhibited similarities across cancer types (i.e., E3, E5, E8, E9, E10, and E13) (Supplementary Fig. S5e), a series of ligand-receptor pairs were identified, which are involved in cancer related pathways, including EGFR (e.g., EGFR_TGFB1), NOTCH (e.g., JAG1_NOTCH2/3), WNT pathways (e.g., FZD6_WNT5A) (Supplementary Fig. S5f).

Given the prolific communication between CAFs and immune cells, particularly $CAF_{State3}$ (e.g., PDCD1_FAM3C interaction in $CAF_{State3}$) (Fig. 3b), it is possible that CAF states may play an important role in checkpoint blockade immunotherapy. For instance, galectin-9 (encoded by *LGALS9*) related interactions have been linked to tolerogenic macrophage programming and adaptive immune suppression[50], and

regulate T-cell death for cancer immunotherapy[51]. Interestingly, a series of interactions between *LGALS9* and its partners were enriched in crosstalk between T-cell/myeloid/endothelial/epithelial and CAF$_{state3}$ but not CAF$_{state1/state2}$ (Fig. 3b, d, f, Supplementary Fig. S5c and f). Therefore, we conducted ssGSEA-based deconvolution analysis to estimate the proportion of CAFs in each state using bulk transcriptome profiles of patients from three independent immunotherapy cohorts with open accessible sequencing data and follow-up information across different cancer types (i.e., melanoma[52], urothelial cancer[53], and uterine sarcoma[54]). Intriguingly, compared to the insignificant associations observed for CAF$_{state1}$ and CAF$_{state2}$ (Supplementary Fig. S6a), a high proportion of CAF$_{state3}$ was significantly associated with poor overall survival in these cohorts ($p = 0.038$, $p = 0.005$, and $p = 0.013$, respectively) (Fig. 3g), even after adjusting available covariates in each cohort study by multivariate analysis ($p = 0.049$, $p < 0.001$, and $p < 0.001$, respectively) (Supplementary Fig. S6b), suggesting the potential independent prognostic value of CAF$_{state3}$ for immunotherapy

## Characterization of the plasticity of fibroblasts via pan-cancer analysis

Fibroblast activation is an important source of mesenchyme-derived stromal components[7]. Except for the main origin of CAFs from NFs (Fig. 2f), CAFs may have alternative origins with various biofunctions in the TME[7]. As described above, three distinct clusters of CAFs with a small number of cells (i.e., c6-c8) were shared by all cancer types (Fig. 1d and Supplementary Data 3). As reported, c8 was defined as antigen-presenting CAFs (CAF$_{ap}$) that overexpressed *ACTA2*, *HLA-DRA*, and *CD74* (Fig. 4a)[55] and was enriched in pancreatic cancer (Supplementary Fig. S7a). Intriguingly, CAF$_{ap}$ presented significantly more interactions with tumor-infiltrating T-cell clusters than CAF$_{myo}$ ($p = 0.002$), with a similar interactive pattern as that of TAM (e.g., LGALS9_HAVCR2/SORL1/CD47) (Fig. 4b). Moreover, CAF$_{ap}$ exhibited a higher transcriptional similarity with mono-macrophage-related clusters (c18-c22) than with other fibroblast clusters (c1-c7) (Supplementary Fig. S7b) and expressed mono-macrophage-specific markers (e.g., *CD68*, *CD163*, and *CD14*) (Fig. 4c). Using the SCENIC analysis described above, several regulons were highly enriched in CAF$_{ap}$ (e.g., *MAFB*, SPI1, and *IKZF1*) (Fig. 2e), which are well-known mediators of polarization and function in TAMs[56–58]. This evidence implied a possible association between CAF$_{ap}$ and TAM.

The transition between macrophages and myofibroblasts has recently been discovered as a pathogenic process that plays a regulatory role in renal fibrosis[59], renal allograft injury[60] and myocardial infarction healing[61]. However, the dynamic alteration between myofibroblasts and macrophages remains unclear. We speculated that CAF$_{ap}$ might be a transitional position between CAFs and TAMs based on previous reports and our findings. With pseudotime trajectory analysis (Fig. 4d), a possible evolutionary TAM-CAF$_{ap}$-CAF$_{myo}$ path was implied. As validation, we found a small population of cells co-expressing α-SMA and CD163 in the stromal compartment of anaplastic thyroid cancer, colorectal cancer and stomach cancer through confocal mIF imaging (Fig. 4d and Supplementary Fig. S7c). Cells with double positivity (α-SMA$^+$ CD163$^+$) ranged from 10.2% to 13.6% of all α-SMA$^+$ cells in these cancer types, which is consistent with our pan-cancer single-cell analysis and a previous report in pancreatic cancer[55]. Given that the regulatory role of CAF$_{ap}$ in Tregs has been experimentally determined in pancreatic cancer[62], the similar functions of this CAF subtype may be shared by different cancer types.

Expression of a series of fibroblast-specific (e.g., *ACTA2* and *MYLK*) and macrophage-specific genes (e.g., *CD163, MAFB*, and SPI1) gradually changed from TAM to CAF$_{myo}$, with CAF$_{ap}$ located at the intermediate position (Fig. 4e). Moreover, CAF- (e.g., *MYLK*) and TAM-specific regulons (e.g., *MAFB* and SPI1) were gradually enriched along the TAM-CAF$_{ap}$-CAF$_{myo}$ trajectory, and CAF$_{ap}$ was present in both regulons

(Fig. 4f), suggesting a the possible transitional position of CAF$_{ap}$ between CAFs and TAMs, which may be an alternative origin of CAFs.

Similarly, c7 was identified as fibroblast-like peripheral nerve cells (CAF$_{PN}$), specifically expressing peripheral nerve-related genes (e.g., *MPZ*, *S100B*, *LGI4*, and *PLP1*) (Fig. 4g). CAF$_{PN}$ was abundant in STAD, COAD/READ, PAAD, PRAD and OV (Supplementary Fig. S7d), in which perineural invasion was considered as an indication of poor prognosis[63]. Additionally, CAF$_{PN}$ enriched unique transcriptional regulons (i.e., *SOX2* and *SOX10*) in regulating myelination of peripheral nerves[64,65] with distinct metabolic hallmarks (e.g., peroxisome, bile acid, and cholesterol homeostasis) (Fig. 2e and Supplementary Fig. S3e).

Besides macrophages and peripheral nerve-derived CAFs, CAFs also exhibited potential plasticity from endothelial cells by endothelial-mesenchymal transition (EndMT), defining c6 as CAF$_{EndMT}$. Intriguingly, CAF$_{EndMT}$ exhibited dual expression of canonical lineage markers of fibroblasts (e.g., *RGS5* and *ACTA2*) and endothelial cells (e.g., *PLVAP* and *VWF*), highlighting the possibility of its involvement in EndMT, which has recently been described in gastric cancer[66] (Fig. 5a). It was confined to all types of tumor/adjacent tissues (Supplementary Fig. S8a) and had more prolific communications with other TME components in tumor/adjacent tissue than normal tissue (Fig. 5b). Moreover, CAF$_{EndMT}$ exhibited dramatic transcriptional similarity with both CAF (e.g., correlation coefficient $R = 0.83$ with CAF$_{myo}$) and TEC ($R = 0.92$) (Fig. 5c). Besides, compared to all the other CAFs and endothelial cells, a unique signature (e.g., *ESM1*) of CAF$_{EndMT}$ (Supplementary Data 9 and Supplementary Fig. S8b) was enriched in ameboidal-type cell/epithelial cell migration signaling (Supplementary Fig. S8c), which was associated with angiogenesis, and consistent with previous hallmark enrichment[67] (Supplementary Fig. S3e). Moreover, CAF$_{EndMT}$ distinctly presented endothelial differentiation- and conversion-related regulons (e.g., *SOX17/18* and *ETS1/2*)[68,69] (Fig. 2e). Consistently, CAF$_{EndMT}$ was located at the transitional position from TEC to CAF$_{myo}$ in the evolutionary trajectory (Fig. 5d). As expected, the angiogenesis hallmark signature (Supplementary Data 7) was highly enriched in CAF$_{EndMT}$, which was significantly higher than both TEC and CAF$_{myo}$ (both $p < 0.001$) (Fig. 5d), suggesting its possible role in angiogenesis. A series of genes were distinctly enriched in each type of cells, and CAF$_{EndMT}$ tended to be involved in several pathways, including ameboidal-type cell migration and vasculature development (Fig. 5e). Moreover, we estimated the CAF$_{EndMT}$ signature in each patient from The Cancer Genome Atlas (TCGA) cohort using ssGSEA-based deconvolution analysis. Due to the small proportion of CAF$_{EndMT}$ in tumors, we compared patients with the top and bottom 20% percentile of the CAF$_{EndMT}$ score. A higher proportion of CAF$_{EndMT}$ is associated with poor prognosis in several types of cancers with large sample sizes, including breast, gastric, and colorectal cancers (Fig. 5f and Supplementary Fig. S9a). Based on the evidence above, CAF$_{EndMT}$ reflects the plasticity of TECs to CAFs and is linked to the initial step of angiogenesis among multiple cancer types, which is associated with metastasis and cancer progression[70].

Collectively, we characterized the shared plasticity of fibroblasts across different cancer types and suggested alternative cell origins of CAFs from TECs at single-cell profiles across cancers.

## The triple interplay between CAFs, TECs, and TAMs in the TME

The triple interplay among fibroblasts, endothelial cells, and myeloid cells was predominant in the TME among all tissue types (Fig. 2b). As stated, increasing interactions between each TME component were observed along normal, adjacent, and tumor tissues ($p < 0.001$). CAF$_{EndMT}$ had the most prolific communications with other TME components, particularly TAM (Supplementary Fig. S9b). Therefore, we speculated that TAMs may play a role in the EndMT process and subsequent tumor angiogenesis. Using a linear regression model, we screened out top 30 genes that were gradually expressed along the

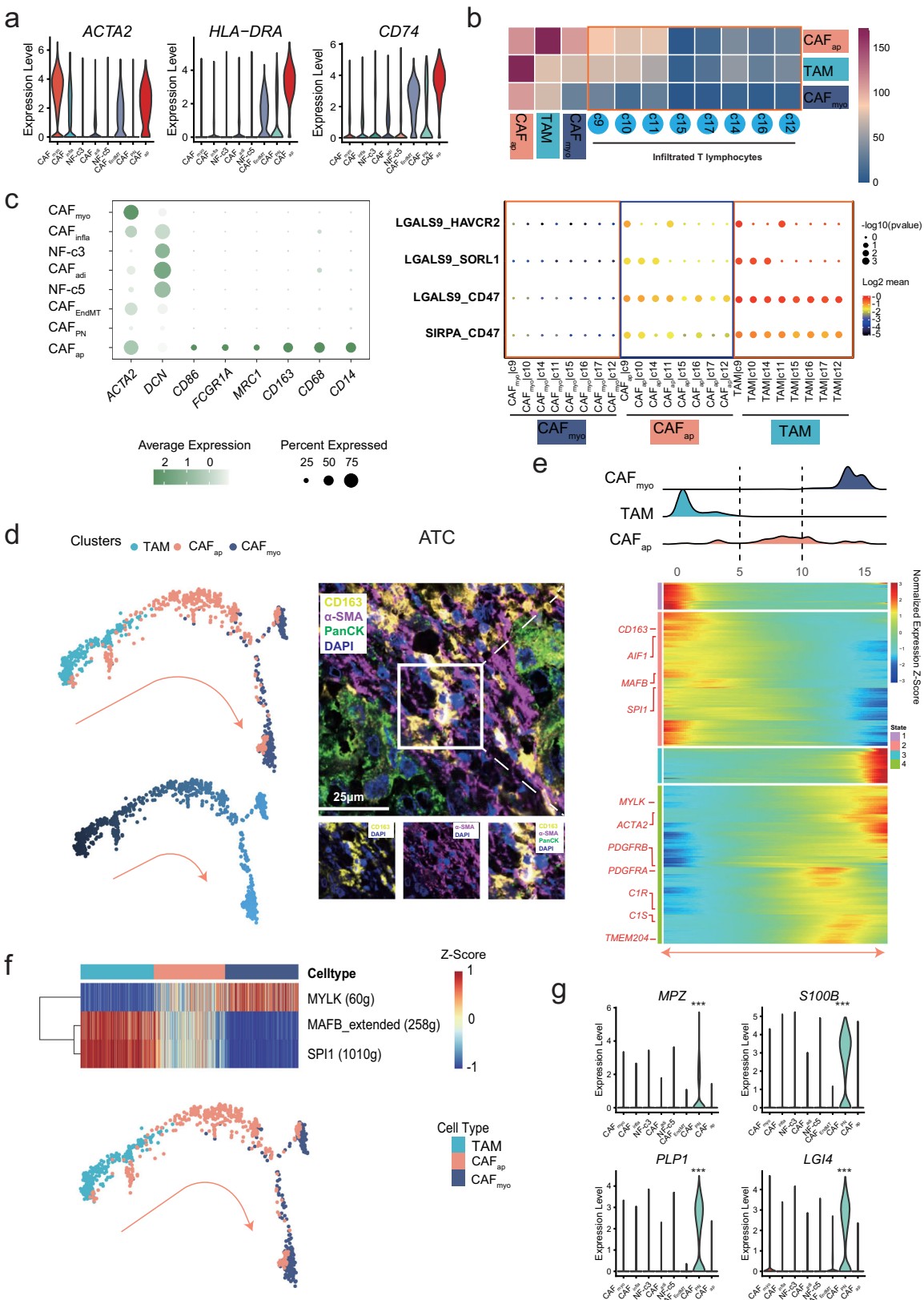

EndMT process from CAF$_{myo}$ to TECs via CAF$_{EndMT}$ (Supplementary Data 10). Intriguingly, *CD44* ranked the top in terms of the interaction counts between CAF$_{EndMT}$ and TAM estimated by NicheNet analysis[71], with SPP1 and APOE exhibited the highest expression in TAM among all its partners (Fig. 6a). Since *APOE* was expressed in nearly 90% of TAMs, we focused on SPP1$^+$ TAMs and found more interactions of CAF$_{EndMT}$

with SPP1$^+$ TAMs than that with SPP1$^-$ TAMs in tumor samples (Fig. 6b). Particularly, the SPP1_a9b1 complex and SPP1_CD44 axes were exclusively enriched in the reciprocal interaction between SPP1$^+$ TAMs and CAF$_{EndMT}$ (Fig. 6c). However, only *CD44* expression increased in a stepwise manner along the EndMT trajectory (Fig. 6d), while no trend was observed for *ITGA9* and *ITGB1* (Supplementary Fig. S9c), whose

**Fig. 4 | Characterization of fibroblast plasticity. a** Violin plot of specific gene expression in antigen-presenting CAFs (CAF$_{ap}$). **b** Upper: peer comparison of the interactions of CAF$_{ap}$, TAM, and CAF$_{myo}$ with T-cell clusters. Lower: the instance of interaction pattern presentation in CAF$_{myo}$, TAM, and CAF$_{ap}$ with T-cell clusters. **c** Bubble plot of mono-macrophage-specific markers in each fibroblast cluster, one-sided Wilcoxon rank-sum test is used to assess the statistical significance of each interaction score. **d** The evolutionary trajectory along the TAM-CAF$_{ap}$-CAF$_{myo}$ path. Confocal image of multiplexed immunofluorescence staining of PanCK, α-SMA,

and CD163 in anaplastic thyroid cancer tissues. Multiplexed immunofluorescence assays are performed twice on tumor samples following assay optimization. **e** Gene expression alteration and ridgeline plot along the reciprocal trajectory. **f** Regulon enrichment along the evolutionary trajectory in different cell types. **g** Violin plot of peripheral nerve-specific genes (*MPZ*, *S100B*, *PLP1*, and *LGI4*) in CAF$_{PN}$. Kruskal-Wallis two-sided test is used to test the significance of gene expression level among different fibroblast clusters. ***$p < 0.001$.

protein products formed the a9b1 complex. This result suggested the possible involvement of the SPP1_CD44 interaction in the EndMT process. We next validated this hypothesis by mIF to assess the spatial distribution of SPP1$^+$ TAMs (SPP1 and CD68) and CAF$_{EndMT}$ (CD44 and CD31, canonical endothelial markers encoded by *PECAM1*, which decline in a stepwise manner during the EndMT process (Fig. 6d)). In three cancer types (i.e., anaplastic thyroid, colorectal, and gastric cancers), mIF consistently illustrated the proximity of some SPP1$^+$ TAMs (SPP1$^+$CD68$^+$) to CAF$_{EndMT}$ (CD44$^+$CD31$^+$) (Fig. 6e). Phenotypic images and density maps were used to quantify marker expression and spatial distribution, respectively (Fig. 6f and Supplementary Fig. S9d). After excluding the defective regions (Supplementary Fig. S9e), the spatial density distribution of CAF$_{EndMT}$ cells was quantified and classified into high-density (HDA) and low-density areas (LDA), and SPP1$^+$ TAMs were significantly enriched in the HDA compared to the LDA (Fig. 6f). Furthermore, when quantifying the spatial distribution, we found that the SPP1$^+$ TAM ratio normalized by the total number of TAMs was significantly higher within 20 µm of CAF$_{EndMT}$ than that outside 20 µm (Fig. 6g). In contrast, the SPP1$^-$ TAM ratio was significantly lower within 20 µm (Fig. 6g). Moreover, a significantly high correlation (R = 0.23, $p < 0.001$) was identified between the signature enrichment of CAF$_{EndMT}$ and SPP1$^+$ TAMs in the spatial transcriptomic profile from seven colorectal tumor samples (Fig. 6h). Additionally, the significantly positive associations of enrichment scores of CAF$_{EndMT}$ and SPP1$^+$ TAMs were also validated in 25 of 28 cancer types in the TCGA dataset (Supplementary Fig. S10), further supporting the possible proximity of SPP1$^+$ TAMs and CAF$_{EndMT}$. The evidence described above suggested that pro-angiogenic SPP1$^+$ TAMs may play a role in the EndMT process to facilitate intratumoral angiogenesis through the SPP1_CD44 interaction between SPP1$^+$ TAMs and CAF$_{EndMT}$ and thus implied poor prognosis in cancer patients. However, mechanistic verifications are largely needed in the future.

## Discussion

Single-cell profiles were investigated in multiple cancer types to reveal the heterogeneity and cancer biology of both cancer and stromal cells. A few studies have conducted pan-cancer analysis to systematically illustrate shared and cancer-type specific characteristics of different cell components, particularly for myeloid and T cells[21,22,24]. Overall, this is a systematic investigation of a single-cell transcriptional atlas of fibroblasts across cancer types.

As an important mesenchyme-derived stromal component, fibroblasts are plastic in phenotype[7]. Pan-cancer analysis offered us an opportunity to characterize the cells in a fluctuating state across various tissues, which was difficult to detect in the original research due to its low population. In the present analysis, the state fluctuation or transition of fibroblasts was associated with multiple biological functions (e.g., angiogenesis, immune modulation, and EMT) and clinical outcomes, particularly after immunotherapy. Although it is crucial to investigate the biological function of CAF subtypes, such as FAP$^+$ CAFs[72], ENG$^+$ CAFs, and THY1$^+$ CAFs, it is more important to explore the generalized characteristics of CAFs in the TME. Here, we found that CAFs had divergent differential states with specific biofunctions. Of note, CAF$_{state3}$, which was at the most dedifferential state, predicted a worse outcome of immunotherapy. Thus, CAF differentiation may promote the stratification of patients with immunotherapy. In

addition, the origin of CAFs was delineated in our study. Consistent with consensus[7], the trajectory of CAFs from state1 to state2/state3 exemplified that most CAFs were likely derived from the activation of local normal fibroblasts. Moreover, the three minor clusters of CAFs (CAF$_{EndMT}$, CAF$_{pn}$ and CAF$_{ap}$) imply alternative origins from endothelial cells, peripheral nerves, and macrophages, respectively.

The *SPP1* gene encodes an integrin-binding glyco-phosphoprotein, named as osteopontin. It is secreted by various tumors and is associated with tumor progression, invasion, and metastasis[73,74]. Although it could be chemokines that recruit macrophages[75], *SPP1* expression was significantly related to TAMs in multiple tumors in the TIMER study[76]. Thus SPP1$^+$TAM should be abounded in tumors. Furthermore, a previous atlas of tumor-infiltrating myeloid cells identified angiogenesis-associated macrophages in 8 cancer types, which was marked by the expression of *SPP1*[21]. However, the potential mechanism was not postulated. In the present study, we found that SPP1$^+$ TAMs may be involved in tumor angiogenesis by interacting with adjacent CAF$_{EndMT}$, which is regarded as the initial step of angiogenesis[70].

Limitations of the study should be noted to avoid over-interpretation. First, some canonical markers may not exhibit restrictive expression in different cell types, thus necessitating expanding evidence to support the findings and avoid possible misleading. Although the expression of *RGS5* and *ACTA2* defined that CAF$_{EndMT}$ was also reported in a recent study[77], the possibility of pericytes could not be excluded. It is well established that pericytes are an important resource of CAFs in tumors[78]. Pericytes recruited by EndMT[78] or pericyte-fibroblast transition[79] lead to tumor vasculature. Overall, pericytes are an important origin of CAFs dependent on EndMT (mainly) or pericyte-fibroblast transition that play roles in angiogenesis[80]. It would be interesting to demarcate or subcluster pericytes from CAFs and determine the difference between them in pan-cancer level in future studies. Second, mechanistic analysis has determined the potential functions of some CAF populations in specific cancer types (e.g., key role of CAF$_{ap}$ on Treg in pancreatic cancer[58]). Although we profiled the ubiquitous characteristics of different CAF subpopulations at single-cell resolution, experimental validation is still warranted to determine whether these functional CAF cells can play a similar role across diverse cancer types. Third, we only demonstrated the possible alternative origin of CAFs mainly through bioinformatic approaches for evolutionary trajectory and illustrated the possible cells at an intermediate state through mIF with specific markers. However, further verification on the dynamics of CAFs and the underlying mechanisms/crucial regulatory factors is lacking, which should be explored in the future.

In conclusion, we systematically characterized CAFs across cancers, not only providing the possible origins of CAFs but also highlighting the different states of CAFs in the outcome of immunotherapy and prognosis. To accelerate the related data mining and application in wide research, we established an interactive website-based tool (https://gist-fgl.github.io/sc-caf-atlas/). Collectively, although further experimental verification is warranted to establish the functional role of each CAF cluster and diverse origins of CAFs, our pan-cancer study on CAFs may facilitate CAF-targeted therapy development and application in the future (Fig. 7).

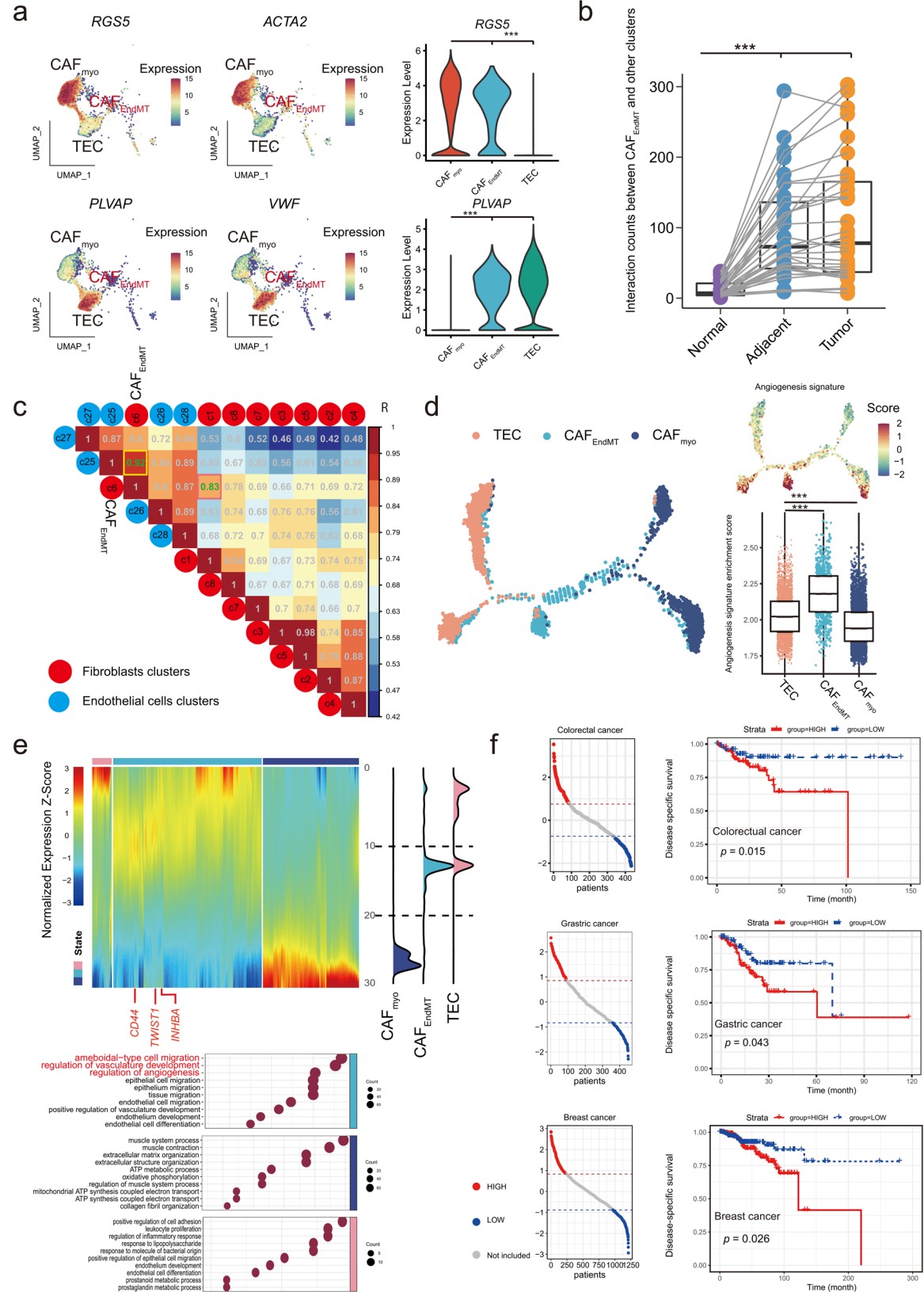

## Methods

### Single-cell RNA-seq data processing

Publicly available and inhouse FASTQ files generated from 10× Genomics were aligned and quantified against the GRCh38 human reference genome using Cell Ranger software (Version 6.1.2) with default settings. The output of the cellranger and count matrix were read using the *Read10X* function from the Seurat package (Version 4.0.4) and *read.table* function, respectively, and the latter was further converted to dgCMatrix format. Potential doublets predicted by Scrublet[81] were removed to avoid interference with the analysis. The *merge* function was used to integrate all individual objects into an aggregate object, and the *RenameCells* function was used to ensure that all cell labels

**Fig. 5 | Characterization of CAFs in endothelial-mesenchymal transition (EndMT). a** Feature and violin plots of specific genes in CAF$_{EndMT}$ and TECs. **b** The interaction counts between CAF$_{EndMT}$ and other components in different tissue origins. Normal: $n = 34$, Adjacent: $n = 34$, Tumor: $n = 34$. The box is bounded by the first and third quartile with a horizontal line at the median and whiskers extend to the maximum and minimum value. Mann–Whitney two-sided test is used to test the significance of interaction counts between CAF$_{EndoMT}$ and other clusters. Normal vs Adjacent $p$-value is $2.52 \times 10^{-6}$, Normal vs Tumor $p$-value is $5.87 \times 10^{-8}$, ***$p < 0.001$. **c** Genetic similarity between clusters of CAFs and endothelial cells. **d** The evolutionary trajectory along the TECs-CAF$_{EndMT}$-CAF$_{myo}$ path with the angiogenesis hallmark signature enriched along the trajectory and CAF$_{EndMT}$. State1: $n = 3550$,

state2: $n = 754$, state3: $n = 3062$. The box is bounded by the first and third quartile with a horizontal line at the median and whiskers extend to the maximum and minimum value. Mann–Whitney two-sided test is used to test the significance of Angiogenesis signatures enrichment scores among TECs-CAF$_{EndMT}$-CAF$_{myo}$ clusters. TEC vs CAF$_{EndoMT}$ $p$-value is $4.67 \times 10^{-102}$. TEC vs CAF$_{myo}$ $p$-value is $5.53 \times 10^{-27}$, ***$p < 0.001$. **e** Gene expression alteration with gene ontology and ridgeline plot along the reciprocal trajectory. **f** Estimation on the prognostic value of the CAF$_{EndMT}$ signature in colorectal, gastric and breast cancer in terms of disease-specific survival. Kaplan–Meier curves for overall survival in all patients according to the number of positive ligands. $p$-values for all survival analyses have been calculated using the log-rank test. Source data are provided as a Source Data file.

were unique. In total, 990,990 cells from different studies were pooled. Furthermore, quality control was applied to the cells based on several criteria. Briefly, cells with <200 detected genes as well as those with >20% mitochondrial content were removed. Cells having over 6000 detected genes were eliminated to further exclude the possible doublets. After filtering, 855,271 high-quality cells were preserved for subsequent analyses. A global-scaling normalization method ("LogNormalize") was employed to ensure that the total gene expression in each cell was equal, and the scale factor was set to 10,000. The top 2000 variably expressed genes were returned for downstream analysis using the *FindVariableFeatures* function. The *ScaleData* function, "vars.to.regress" option UMI, and percent mitochondrial content were used to regress out unwanted sources of variation. Principal component analysis (PCA) incorporating highly variable features reduced the dimensionality of this dataset, and the first 30 PCs were identified for analysis. To remove batch effects, the *RunFastMNN* function in SeuratWrappers package (Version 0.3.0) was selected to perform sample batch correction. Clustering analysis was performed based on the edge weights between any two cells, and a shared nearest-neighbor graph was produced using the Louvain algorithm, which was implanted in the *FindNeighbors* and *FindClusters* functions. The identified clusters were visualized using the UMAP method. For subclustering analysis, a similar procedure was applied, including normalization, variably expressed feature selection, dimension reduction, batch correction with *RunFastMNN*, and clustering identification. To annotate the cell clusters, differentially expressed markers of the resulting clusters were identified with the *FindAllMarkers* function using the default nonparametric Wilcoxon rank sum test with Bonferroni correction.

## Comparison dendrograms

To demonstrate that the subpopulations of CAFs or myeloid cells were not heterogeneous across tumor types, an unsupervised comparison dendrogram was performed. We selected the top 2000 highly variable genes across different subclusters. The mean expression of these genes in each cluster was used to calculate the Pearson correlation coefficient with the psych package (Version 2.2.5). The distance defined as (1-Pearson correlation coefficient)/2 was adopted for hierarchical clustering. For visualization, the factoextra package (Version 1.0.7) was applied.

## Cell–cell interaction analysis

CellphonedDB[19] was used to analyze cell–cell interactions among all TME components. Input files for the *statistical analysis* function comprised a raw count matrix extracted from the Seurat object and an annotation file of cell types. *The heatmap_plot* function from CellphoneDB and the Circlize package (Version 0.4.14) were used to display the frequency of interactions between two cell subsets. Visualization of the potential interaction strength between ligand and receptor, which was predicted based on their average expression, was performed using the *dot_plot* function and pheatmap package (Version 1.0.12). Significant ligand–receptor pairs ($p < 0.01$) were extracted for illustration.

## Trajectory analysis

To investigate dynamic biological processes, such as interconversion and evolutionary trajectories of different cell types, we applied the Monocle (Version 2.22.0) algorithm[82]. The *NewCellDataSet* function was used to create a new object for the monocle using transcript count data of the included cell populations. The results generated from *estimateSizeFactors* and *estimateDispersions* function assisted us in normalizing for differences in mRNA recovered across cells and performing differential expression analysis later. Signature genes expressed in at least 10% cells of the dataset and with a $p < 0.01$ calculated using the *differentialGeneTest* function were included to define the trajectory progress. The *ReduceDimension* function reduced the space down to two dimensions, and the *orderCells* function ordered the cells according to gene expression. Pseudotime-dependent genes were calculated using *differentialGeneTest* and the "fullModelFormulaStr" option "-sm.ns(Pseudotime)", and smooth expression curves were generated with the *plot_pseudotime_heatmap* function. The Ggridges package (Version 0.5.3) was used to analyze the frequency of distributed cells in different groups on the pseudotime axis.

## Enrichment analysis

Pseudotime-dependent genes were further subjected to GO and KEGG enrichment analysis using the clusterProfiler package (Version 3.0.4) with default settings. Fifty hallmark gene sets in the MSigDB database (https://www.gsea-msigdb.org/gsea/msigdb) were used for GSEA of clusters c6/c7/c8 with the escape package (Version 1.4.0). A nonparametric and unsupervised algorithm from the gene set variation analysis (GSVA) package (Version 1.14.1) was selected to assess the EMT and ANGIOGENESIS scoring of different states generated with Monocle. The signature genes of EMT and ANGIOGENESIS were obtained from fifty hallmark gene sets.

## Single-cell regulatory network inference and clustering (SCENIC) analysis

Cells in different states confirmed using Monocle were further included in the SCENIC package (Version 1.2.4)[83], and they were then sorted based on clusters and states. To remove noise, genes with low expression levels or low positive rates were filtered using the *geneFiltering* function with default settings. Additionally, only the genes that matched the Rcis target databases were retained for downstream analysis. After reconstruction of the gene regulatory network, GENIE3 detected the relationship between transcription factors and potential targets. A total of 24,453 motifs from the cisTarget Human motif database v9 were used for enrichment of gene signatures, which were pruned for targets according to cis-regulatory cues using default settings. The enrichment of regulons across single cells was identified using the "aucell" positional argument, and the results were visualized using the pheatmap package (Version 1.0.12).

## NicheNet analysis

NicheNet (Version 1.0.0)[71] was used to identify potential ligands that drive the phenotype of cluster c6. The top 20 differentially expressed genes were ordered using log2FC between cluster c6,

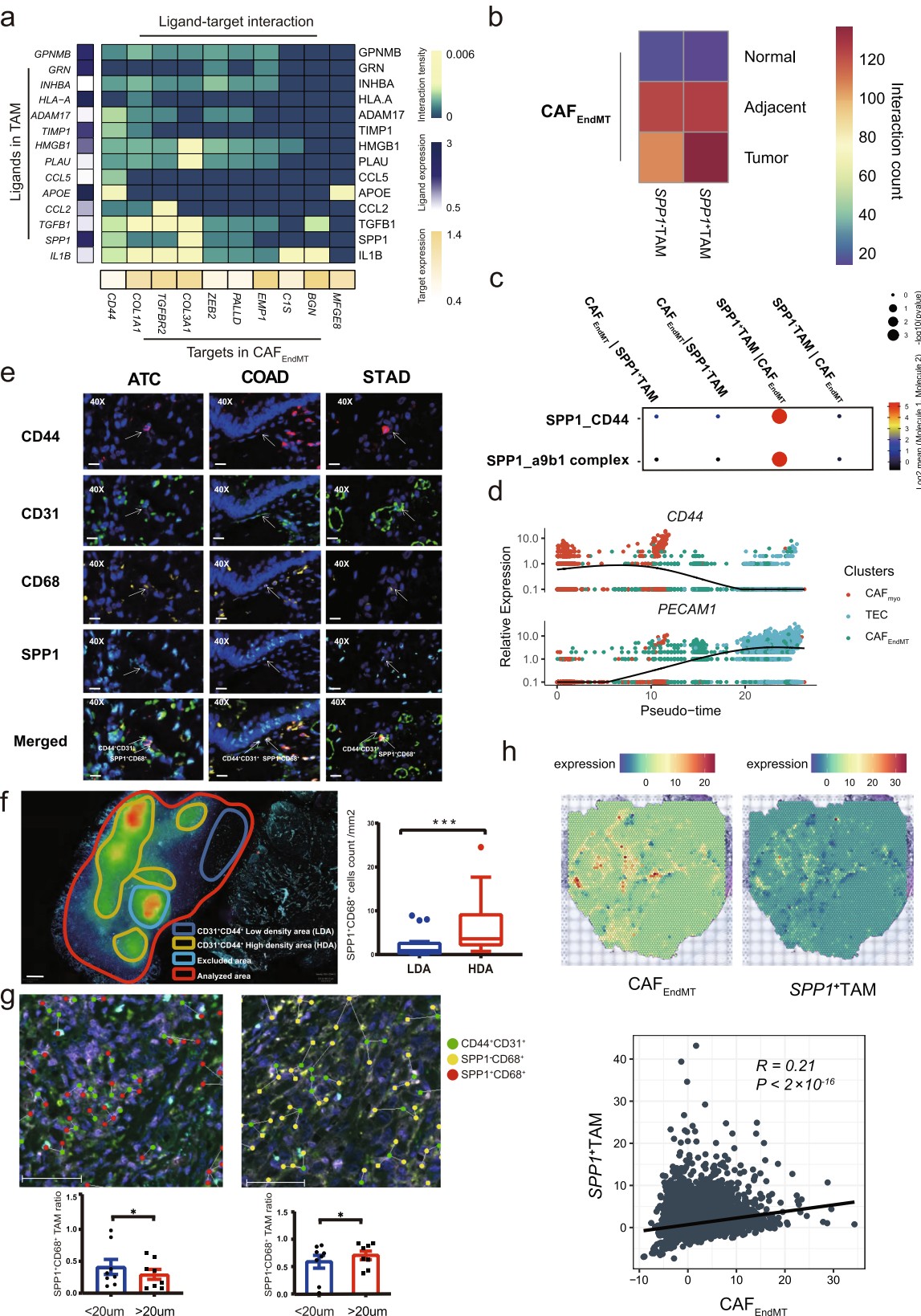

and the remaining clusters were treated as potential targets. Genes with a positive rate of >10% in clusters c6 and c19 were considered background and potential ligands, respectively. Sender cells from c19 and target cells from c6 were included to construct the expressed ligand–receptor interactions and calculate the ligand activity. We used the *active_ligand_target_links* function to compute the potential intensity of regulation between the ligand and target.

## Similarity analysis

To quantify the similarity among different subclusters of fibroblasts and endothelial cells, the top 5000 variably expressed genes were included

**Fig. 6 | Triple interplay between CAF_EndMT and SPP1⁺ TAMs. a** NicheNet analysis screening potential ligands of CD44. The left bar presents the expression scale of the potential ligand in TAM. **b** Predicted interaction counts between CAF_EndMT and SPP1⁺ TAM/SPP1⁻ TAM using CellphoneDB analysis. **c** SPP1-involved specific ligand–receptor interaction between CAF_EndMT and SPP1⁺/SPP1⁻ TAMs, one-sided Wilcoxon rank-sum test is used to assess the statistical significance of each interaction score. **d** Dynamic alterations in *CD44* and *PECAM1* during EndMT. **e** Multiplexed immunofluorescence staining of CD44, CD31, CD68, and SPP1 in anaplastic thyroid cancer, gastric cancer, and colorectal cancer tissues, Scale bar: 20 μm. Multiplexed immunofluorescence assays are performed twice on tumor samples following assay optimization. **f** Illustration of CD44⁺CD31⁺ high- and low-density areas (HDA and LDA, respectively) and the quantified results (LDA: *n* = 23, HDA: *n* = 17, The box is bounded by the first and third quartile with a horizontal line at the median and whiskers extend to the maximum and minimum value. Mann–Whitney two-sided test is used to test the

significance of proportion between LDA and HDA categories. ***\*p < 0.001, p-value is 0.0008), Scale bar: 500 μm Multiplexed immunofluorescence assays are performed twice on tumor samples following assay optimization. **g** The spatial distance quantification. The left panel compares the SPP1⁺CD68⁺ and the right panel compares SPP1⁻CD68⁺ macrophage ratios (normalized by the total number of macrophage) between within 20 μm and outside 20 μm of CAF_EndMT (<20 μm: *n* = 8, >20 μm: *n* = 8, Wilcoxon two-sided test is used to test the significance of ratio between within 20 μm and outside 20 μm of CAF_EndMT. *\*p < 0.05, left panel: p-value is 0.0391, right panel: p-value is 0.0391). Scale bar: 50 μm. **h** Upper: Illustration of the spatial transcriptomic spot of colorectal cancer tissues with CAF_EndMT and SPP1⁺TAM signature enrichment. Lower: The scatter plot and correlation between the CAF_EndMT enrichment score and SPP1⁺TAM enrichment score (R represents Pearson's correlation and its coefficient of determination, p-value is 2.26 × 10⁻¹⁰), suggesting the co-localization of these two cell types. Source data are provided as a Source Data file.

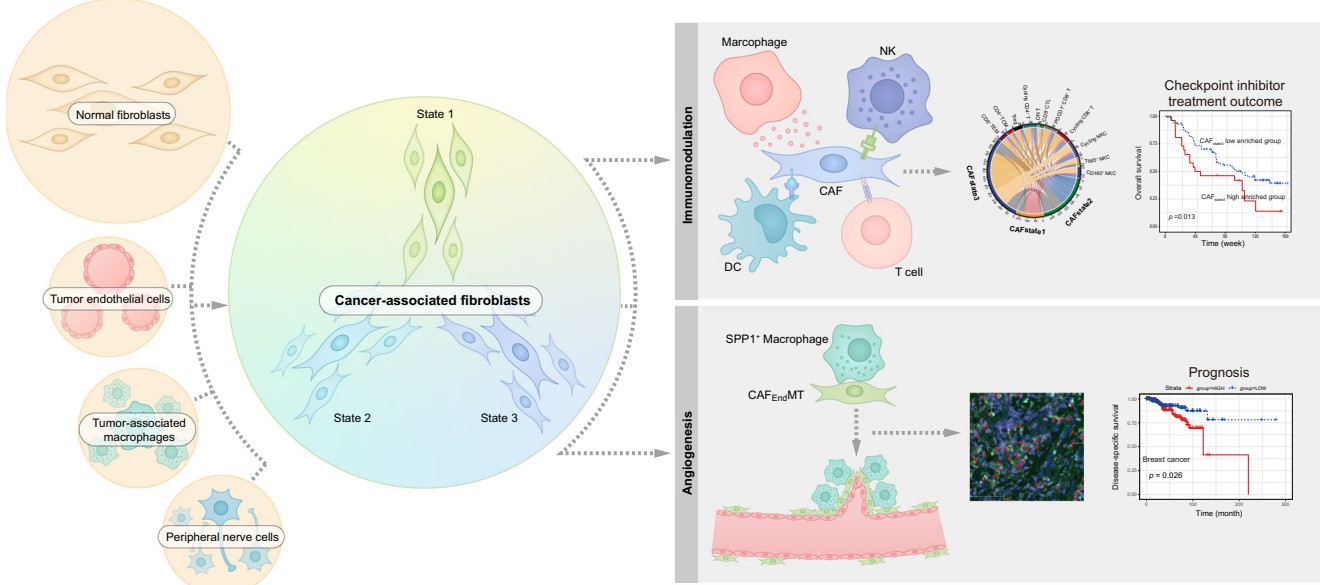

**Fig. 7 | Summary illustration of the study.** Left part: four circles stand for the origins of cancer associated fibroblast (CAF). The big circle indicates the main origin of CAF-derived from normal fibroblasts activation, whereas the three small circles indicate the alternative origin of CAF. In general, the activation trajectory is divided into three states (state 1–3). Right part: The state of CAF is associated with immunomodulation, thus may predicting the prognosis of checkpoint-inhibitor-based treatment for specific cancer types, and it is also associated with angiogenesis by interacting with proximal SPP1⁺ macrophages and prognosis of cancer patients.

in the *corr.test* function from the psych package (Version 2.2.5), and the corrplot package (Version 0.92) was used for visualization.

### Survival analysis

mRNA expression counts and clinical information from TCGA[84] were downloaded from Firebrowse (http://firebrowse.org/). Based on the enrichment scores of CAF_EnoMT signature gene sets (SPP1, CD44, CD68, PECAM1, VCAN, CD14, and MARCO) calculated with the GSVA package, patients with top and bottom 20% scores were selected for subsequent survival analysis. CIBERSORT[85] was used to infer the abundance of each CAF state from available bulk RNA-seq data in four immunotherapy cohorts with detailed follow-up information. TPM or TMM values were used because their results show small root-mean-square error (RMSE) and high Pearson correlation values. B-mode was used to remove the batch effect. The algorithm was run with a web portal autogenerated signature matrix and 1000 permutations. Survival (Version 2.42–3) and Survminer (Version 0.4.9) packages were used for analysis and visualization.

### Spatial transcriptomics

Raw base call files were converted to FASTQ reads using bcl2fastq. Reads were mapped to the human reference genome GRCh38-2020 using Space Ranger (Version 1.3.1) software[86], and 332 million high-

quality uniquely mapped reads were obtained. The median numbers of reads and genes detected per spot were 98,830 and 1,803, respectively. Read10X_h5 and the *CreateSeuratObject* function from the Seurat package were used to create an object with the output of Space Ranger. With the help of the Read10X_Image function, we loaded the H&E image data and used a standard *logNormalize* function to normalize the dataset. The *SpatialFeaturePlot* function showed the expression level of a single gene at a spatial location. Spots where the expression of the four genes (*CD44*, SPP1, *PECAM1*, and *CD68*) was simultaneously nonzero were recolored, and the color intensity reflected the average expression level of the signature set. We selected the *corr.test* function from the psych package (Version 2.2.5) to calculate the correlation of genes belonging to the signature set.

### Multiplexed immunofluorescence (mIF) analysis

All involved surgical tissue samples were processed into paraffin blocks and cut into 5-μm-thick FFPE sections. Multiplex IHC staining was performed using an Opal 7-color kit (Akoya Bioscience, NEL801001KT). The relative markers CD68 (ab213363, Abcam, 1:1000, Opal 620), SPP1 (ab214050, Abcam, 1:1000, Opal 520), CD31 (ab182981, Abcam, 1:2000, Opal 480), and CD44 (ab213363, Abcam, 1:2000, Opal 690) were evaluated via IHC. Briefly, the sections were dewaxed with xylene for 20 min, and ethanol was used for rehydration. Microwave

treatment was performed for antigen retrieval with buffer (pH 9.0). Next, all sections were cooled for 30 min to room temperature. Endogenous peroxidase activity was blocked using an antibody diluent/block (72424205; Akoya Bioscience) at room temperature for 10 min. Slides were incubated with a primary antibody at room temperature for 1 h, followed by secondary reagents at 37 °C for 20 min and tyramide signal amplification reagents at room temperature for 10 min (Opal 480, Opal 520, Opal 620, and Opal 690, Akoya Bioscience, 1:100). MWT antigen retrieval was performed until all markers were stained. Nuclear staining was performed using DAPI (Akoya Bioscience, 1:5) at room temperature for 5 min. Slides were mounted using anti-Fade fluorescence mounting medium (ab104135, Abcam), and they were stored at 4 °C until image acquisition. Slides were scanned using a PerkinElmer Vectra Polaris (PerkinElmer) and a confocal microscope (TCS SP8, Leica). The percentage of positively stained cells among all nucleated cells was determined. Multispectral image unmixing was performed using QuPath software (version 3.0)[87] and ImageJ (version 1.53i). Briefly, DAPI-positive cells were identified using the "cell detection" command, and each single channel threshold was selected. Following this, all detected cells were divided into different subgroups for further analysis, and defective samples or areas with staining artifacts were reanalyzed or excluded. The in-house data were obtained from Novogene Co., Ltd.

### Reporting summary
Further information on research design is available in the Nature Research Reporting Summary linked to this article.

## Data availability
All the expression data can be obtained from the Gene Expression Omnibus, and the selected studies are listed in Supplementary Data 1. Analysis and visualization of the scRNA-seq datasets in this study can also be performed at https://gist-fgl.github.io/sc-caf-atlas/. Additionally, the integrated single-cell RNA sequencing matrix data that support the findings of this study are deposited in Gene expression Omnibus (accession No. GSE210347). Previously published scRNA-seq data reanalyzed here are available under accession codes GSE134355 (Normal data by Han et al.[25]), GSE141445 (Prostate cancer data by Chen et al.[88]), E-MTAB-8107 [https://www.ebi.ac.uk/biostudies/arrayexpress/studies/E-MTAB-8107/sdrf] (Ovary/Breast/Colorectum cancer data by Qian et al.[24]), GSE157703 (Prostate cancer data by Ma et al.[89]), GSE131907 (Lung cancer data by Kim et al.[26]), GSE138709 (Intrahepatic cholangiocarcinoma data by Zhang et al.[90]), E-MTAB-6149 [https://www.ebi.ac.uk/biostudies/arrayexpress/studies/E-MTAB-6149/sdrf], E-MTAB-6653 [https://www.ebi.ac.uk/biostudies/arrayexpress/studies/E-MTAB-6653/sdrf] (Lung carcinomas data by Qian et al.[24]), CRA001160 (PDAC data by Peng et al.[91]), GSE154778 (PDAC data by Lin et al.[92]), HRA000212 (Bladder cancer data by Chen et al.[93]), HRA000686 (Thyroid cancer data by Luo et al.[39]). The gastric cancer data by Sathe et al.[94] were downloaded from [http://dna-discovery.stanford.edu/download/1401/]. The raw sequencing data for the spatial transcriptome from this study have been deposited in the Genome Sequence Archive in BIG Data Center, Beijing Institute of Genomics, Chinese Academy of Sciences, under accession numbers (HRA003299 and HRA003300) that can be accessed at https://ngdc.cncb.ac.cn/gsa-human/. Source data are provided with this paper.

## Code availability
The bioinformatic analysis code has been uploaded into the github (https://github.com/Xiaxy-XuLab/PanCAF, https://doi.org/10.5281/zenodo.7095147)[95].

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

## Acknowledgements
Appreciates the assistance of Mis. Xinyue Hu in illustration (Fig. 7) design and drawing. This study was supported by the National Key R&D Program of China (No. 2021YFA1301203(H.X.), 2018YFC2000305(L.Z.D.)), National Natural Science Foundation of China (82103031(H.L.), 81973408(H.X.), 82272933 (H.L.), 82273445(Y.S.)), International Cooperation Project of Chengdu Municipal Science and Technology Bureau (2020-GH02-00017-HZ(H.L.)), 1.3.5 Project for Disciplines of Excellence, West China Hospital, Sichuan University (No. ZYYC20003(H.X.), ZYJC18035(H.L.), ZYJC18025(Z.H.L.), ZYJC18003(Y.L.), ZYJC18004(H.X.), and ZYYC20007(L.D.)), and the National Research Foundation of Korea funded by the Korean government (2019R1C1C1005403(J.P), 2019R1A4A1028802(J.P), 2021M3H9A2097520(J.P)), Clinical Research Incubation Project, West China Hospital, Sichuan University (22HXFH019(H.L.)).

## Author contributions
H.L., J.P., and H.X. designed and supervised the project. X.Y.X., M.Y.C., Y.S., Z.X.R., K.D and P.-H.L. analyzed the data. H.L., H.R.T., B.B., W.G.J., Yi.L., W.Z., L.Y., Y.P., L.Z.D., H.B.H., Y.G.H., J.Q.Z., J.P., C.C., and H.X., interpreted the data. H.L., H.-N.C., W.-H.Z., Yang.L., R.H.S., C.L., and Z.H.L. collected the clinical samples and information. L.-B.H., X.Y.K., and H.J. performed multiplexed immunofluorescence staining and conducted related analysis. H.A. and J.P. built the website. Y.J. conducted pathological evaluation. X.Y.K. draw the diagrams. H.L., J.P., and H.X. wrote and revised the manuscript.

## Competing interests
The authors declare no competing interests.

## Additional information

Han Luo [1,2,3,4,18] ✉, Xuyang Xia[4,5,18], Li-Bin Huang [3,6,18], Hyunsu An [7], Minyuan Cao[5], Gyeong Dae Kim [7], Hai-Ning Chen [3,8], Wei-Han Zhang[3,9], Yang Shu [5,9], Xiangyu Kong[1,2,3], Zhixiang Ren[5], Pei-Heng Li [1,2,3], Yang Liu[1,2,3], Huairong Tang[10], Ronghao Sun[1,2,11], Chao Li[11], Bing Bai[12], Weiguo Jia[13], Yi Liu[14], Wei Zhang[15], Li Yang[5], Yong Peng [5], Lunzhi Dai [5], Hongbo Hu [5], Yong Jiang[16], Yiguo Hu [1,5], Jingqiang Zhu[1,2], Hong Jiang[5], Zhihui Li [1,2], Carlos Caulin[17], Jihwan Park [7] ✉ & Heng Xu [4,5] ✉

[1]Division of Thyroid and Parathyroid Surgery, West China Hospital, Sichuan University, Chengdu, Sichuan, China. [2]Laboratory of thyroid and parathyroid disease, Frontiers Science Center for Disease-related Molecular Network, West China Hospital, Sichuan University, Chengdu, Sichuan, China. [3]Department of General Surgery, West China Hospital, Sichuan University, Chengdu, Sichuan, China. [4]Division of Laboratory Medicine/Research Centre of Clinical Laboratory Medicine, West China Hospital, Sichuan University, Chengdu, Sichuan, China. [5]State Key Laboratory of Biotherapy and Cancer Center, West China Hospital, Sichuan University, Chengdu, Sichuan, China. [6]Division of Gastrointestinal Surgery, State Key Laboratory of Biotherapy, West China Hospital, Sichuan University, Chengdu, Sichuan, China. [7]School of Life Sciences, Gwangju Institute of Science and Technology (GIST), Gwangju, Republic of Korea. [8]Colorectal Cancer Center, West China Hospital, Sichuan University, Chengdu, Sichuan, China. [9]Gastric Cancer Center, West China Hospital, Sichuan University, Chengdu, Sichuan, China. [10]Health Promotion Center, West China Hospital, Sichuan University, Chengdu, Sichuan, China. [11]Department of Head and Neck Surgery, Sichuan Cancer Hospital and Institute, Sichuan Cancer Center, School of Medicine, University of Electronic Science and Technology of China, Chengdu, Sichuan, China. [12]State Key Laboratory of Primate Biomedical Research, Institute of Primate Translational Medicine, Kunming University of Science and Technology; Yunnan Key Laboratory of Primate Biomedical Research, Kunming, Yunnan, China. [13]Center for Geriatrics medicine, West China Hospital, Sichuan University, Chengdu, Sichuan, China. [14]Division of Rheumatism & Immunology, Rare Diseases Center, West Chia Hospital, Sichuan University, Chengdu, Sichuan, China. [15]Department of Clinical Pharmacology, Hunan Key Laboratory of Pharmacogenetics, Xiangya Hospital, Central South University, Changsha, Hunan, China. [16]Division of Pathology, West China Hospital, Sichuan University, Chengdu, Sichuan, China. [17]Department of Otolaryngology - Head & Neck Surgery and University of Arizona Cancer Center, University of Arizona, Tucson, AZ, USA. [18]These authors contributed equally: Han Luo, Xuyang Xia, Li-Bin Huang. ✉e-mail: luohan-hx@scu.edu.cn; jihwan.park@gist.ac.kr; xuheng81916@scu.edu.cn

