## [Peer review file · Nature Communications]

REVIEWER COMMENTS

Reviewer #1 (Remarks to the Author): Expert in single-cell RNA-seq, tumour microenvironment, and cancer genomics

Reviewer comment

In the manuscript entitled “Pan-cancer single-cell analysis reveals the key role of cancer-associated fibroblasts in the tumor microenvironment”, the authors performed an integrated analysis on the public single cell RNA sequencing datasets across 10 cancer types to identify the diversity of cancer associated fibroblasts (CAFs) and their plastic/prognostic features in relation to other cell types populating the tumor microenvironment. This is the largest scale and extensive integrated analysis focusing on CAFs and well-suited to demonstrate similarities and differences in CAFs for diverse cancer types.

Despite the scale and meticulousness, I find a weakness on the data composition and integrated analysis strategies. The study was performed on 10 cancer types yet normal or adjacent cells are mostly provided by 4 cancer types. The study heavily utilized trajectory analysis to suggest plasticity of CAFs, which may not be drawn from a single cancer type. Normal lung fibroblasts are not expected to go to colon cancer sites. The cellular interaction or gene network analysis performed on the integrated datasets suffer from the same flaw. Finding similarities and diversities benefits from the integrated analysis of pan-cancer, but inferring plasticity or interaction does not. In addition, most findings are speculative and supporting data (mIF) are not convincing.

Major comment

1. Page 7, lines 177-180: According to the original paper, early stage tumors are from surgical resection and late stage tumors from biopsies; many from lymph node biopsies and very few fibroblasts were recovered. Brain metastasis samples are resected from the brain. Due to the differences in the sampling sites and methods, the statement is over-interpretation.
2. Page 8, lines 202-207 related to Supplementary Fig 1g: A single picture of co-localization is not very convincing. What proportion of CD11c+CD86+ populations co-express CD3? Providing flow cytometry data or confocal images will be better to exclude adjacent cells of CD11c+CD86+ and CD3+.
3. Lines 208-213 related to supplementary figure S1h: Cluster5 rarely express B cell markers and looks like mis-classified cells. Please provide cluster-level differentially expressed genes (DEGs) and check whether they are misclassified T cells or potential doublets.
4. Figure 2a and b, Figure 3: Integrated pan-cancer analysis for cellular interaction is odd and misleading.

5. Figure 2f-h, Figure 4d: Trajectory may be used the gene expression comparisons shared between different cancer types. However, integrated pan-cancer trajectory analysis is odd and misleading for the inference of cellular transformation.
6. Figure 4d: A single picture of co-localization is not very convincing. What proportion of α -SMA+ populations co-express CD163? Providing flow cytometry data or number of confocal images will be more convincing.
7. Figure 5f: Is it possible to distinguish between signatures of EndMT and of abundant CAFs and endothelial cells? If possible, please provide the unique EndMT signature not detected in CAFs or endothelial cells.
8. Line 434-436 related to Figure 6h: the resolution of spatial transcriptomics is ~ 10 cells and does not allow distinction of co-localization of two cell types from a single cell type with co-expression of the genes. It should be clearly noted in the text to avoid over-estimation of the data.

Reviewer #2 (Remarks to the Author): Expert in single-cell RNA-seq and computational cancer genomics

The manuscript presents a pan-cancer single cell analysis of the tumor microenvironment, with a focus on fibroblasts. The authors have a great grasp of vexing issues around fibroblasts. They provided an excellent framing of these issues in the introduction, particularly on topics of fibroblast plasticity and interactions in TME.

The main strength of this study is the work to harmonize multiple scRNAseq data sets across different cancer and provide pancancer clusters. This harmonized data will be of interest to the broader community and likely regarded as an important resource.

The authors use the harmonized scRNAseq data for exploration and discovery with a focus on the nonmalignant compartment of the tumor microenvironment (TME). The analyses are interesting but mostly descriptive (i.e. proportion of cell types under different conditions, expression of canonical markers), and lack statistical rigor. The authors have the sample size to provide statistical significance but few p-values are reported. Adding statistical significance to all claims is needed for this study.

The authors make at least three scientific claims that are backed up to varying degrees. First, the authors claim to identify new cell types, namely CD3+ DC and Bregs. Claiming new cell types from scRNAseq data is very tricky business. Limited IF images are provided for the CD3+ DC and nothing for the Bregs. I recommend these speculative components are moved to the supplement.

Second, the authors claim that they have identified a fibroblast cell state that is associated with immunotherapy response. I found this part of the work most compelling, particularly the survival association. For a more convincing result, the author need to determine if the fibroblast cell type of interest added prognostic significance independent of other covariates. If so, I recommend that the authors provide richer insights (likely cell-cell interactions?) to support this association.

Third, the authors consider CAF transdifferentiation through ENDOMT and the TAM plasticity. This work is intriguing but not presented in a statistically rigorous manner. Here the authors a propose several hypotheses but do not frame their analysis as a hypothesis test to make this aspect of their work convincing. One idea to consider to create null distributions from all the cell types not likely to be associated with transdifferentiation and derive a p-value for the findings reported.

Throughout the entire manuscript claims of casuality are substantially overstated. There is no mechanistic work in this study so the authors can not make claim that, for example, a specific subset of fibroblasts “sculpts” the TME. Claims of casuality from presumed cell-cell interactions and pseudo time trajectories are misleading without any mechanistic work.

A major missing component of this work is the interaction of non-malignant cells with malignant cells in the TME. There is surprisingly no consideration tumor-fibroblast interactions. Moreover, the authors considerations of cell-cell interactions in a pancancer context is limited. There is little exploration findings that are shared or not shared across different cancer types. Exploration of fibroblast-malignant interactions would strengthen the study, especially if reported in a pancancer context.

Reviewer #3 (Remarks to the Author): Expert in cancer-associated fibroblasts, tumour microenvironment, and cell plasticity

In this manuscript, Luo et al. analyze normal fibroblasts and CAFs across multiple tissue/cancer types. Three distinct differentiated states of CAFs were identified with distinct associations with response to immunotherapy. In addition, the authors propose that CAFs arise from macrophages, peripheral nerve, and endothelial cells. While this manuscript provides a comprehensive analysis of CAFs across cancer types, the evidence for the origins of CAFs and their presumed functions is lacking. Several points should be addressed, as outlined below.

1. The authors identify a subset of regulatory B cells that express CD3D, IL32, and TIGIT (cluster 5); however, this cluster expresses low levels of CD79A and MS4A1, suggesting that this subset may be misclassified as B cells. Expression of CD79A and MS4A1 in T cell subsets should be included to support the argument that this cluster is in fact B cells.
2. The expression of SPP1 and C1QC shown in in Figure 1 does not support the subclustering of SPP1+ TAMs and C1QC TAMs, due to the extensive overlap and poor segregation in the current setting. Also, all the other TAM markers were not shown.
3. The authors state that the distribution of fibroblast clusters is altered in tumor and adjacent tissue compared to normal tissue, but it is difficult to appreciate the relative distribution of fibroblasts based on the way the data is currently presented. Including the clustering of fibroblasts in normal, tumor, and adjacent tissues would clarify differences in the tissue types and also how normal fibroblasts were defined (i.e., if normal fibroblasts are a largely distinct subset from adjacent and tumor-associated fibroblasts). Specifically, in Figure 2, what are the defining (unique) marker genes of NFs, in contrast to CAFs?
4. The similarities and differences of CAFs (and CAF subpopulations) across various cancer types appear to be overlooked. Is the CAF-state (state 1, state 2, state 3) universal to various cancer types?
5. Some of the fibroblast clusters identified express high levels of ACTA2 and RGS5 (cluster 1 and CAFEndoMT) which the authors refer to as canonical fibroblast markers. While ACTA2 and RGS5 mark fibroblasts, they are also expressed in perivascular smooth muscle cells/pericytes and the cells detected in these subsets may not be fibroblasts derived from EndMT. Consequently, this caveat should be included in the manuscript or additional evidence that these subsets are not pericytes provided.
6. PDGFRA, PDGFRB, FAP, NOTCH3, HES4, and THY1 were enriched in CAFs compared to NFs, which the authors attribute to differences in functions related to angiogenesis and immunomodulation, but the majority of these genes are CAF biomarkers (PDGFRA, PDGFRB, FAP, and THY1). Similarly, genes highly expressed in CAFstate2 (PDGFRB) and CAFstate3 (PDGFRA, ENG, and FAP) are CAF markers. As a result, increased expression is not necessarily related to functions in angiogenesis or immunomodulation, rather an indication of increased activation into CAFs. The conclusions about potential functions should be revised or additional non-CAF biomarker genes included to support this hypothesis.
7. In the same vein, the correlation between CAFs and TAMs is only descriptive, without any functional validation. This should be revised to avoid over interpretation of the results.
8. In Figure 5, it is unclear why the EC and CAF clusters are closely adjacent, in contrast to all other studies showing clear, distinct segregation between these two clusters. This raises questions with respect to the data analysis methodology of this study. Where are EC and CAF clusters located in Figure 1C? Also, representative EC and CAF marker genes are missing here. The CAF-endMT conclusion is based on descriptive data in Figure 5, without any functional validation. This should be revised to avoid over interpretation of the results.
9. In addition, can the author clarify whether the correlation between CAF-endMT and SPP1 TAM is applicable to other cancer types in clinical datasets such as TCGA?

10. The relevance of the EMT score to the trajectory of CAFs is unclear given CAFs do not have prominent epithelial feature to begin with. Please clarify.

Minor comments:

1. In Fig. 4D, it would help to include split channel images to better visualize the overlap between CD163 and aSMA.
2. The phenotype image in Fig. S53 does not appear to match composite image.
3. 'EndMT' is more often used than 'EndoMT' and the nomenclature used by those who initially reported this phenomenon.

REVIEWER COMMENTS

Reviewer #1 (Remarks to the Author): Expert in single-cell RNA-seq, tumour microenvironment, and cancer genomics

Reviewer comment

In the manuscript entitled “Pan-cancer single-cell analysis reveals the key role of cancer-associated fibroblasts in the tumor microenvironment”, the authors performed an integrated analysis on the public single cell RNA sequencing datasets across 10 cancer types to identify the diversity of cancer associated fibroblasts (CAFs) and their plastic/prognostic features in relation to other cell types populating the tumor microenvironment. This is the largest scale and extensive integrated analysis focusing on CAFs and well-suited to demonstrate similarities and differences in CAFs for diverse cancer types.

Despite the scale and meticulousness, I find a weakness on the data composition and integrated analysis strategies. The study was performed on 10 cancer types yet normal or adjacent cells are mostly provided by 4 cancer types. The study heavily utilized trajectory analysis to suggest plasticity of CAFs, which may not be drawn from a single cancer type. Normal lung fibroblasts are not expected to go to colon cancer sites. The cellular interaction or gene network analysis performed on the integrated datasets suffer from the same flaw. Finding similarities and diversities benefits from the integrated analysis of pan-cancer, but inferring plasticity or interaction does not. In addition, most findings are speculative and supporting data (mIF) are not convincing.

Response: We greatly appreciate these constructive comments and suggestions. We totally agree that the absolute number of normal and adjacent cell are mostly provided by 4 cancer types due to the data availability. Indeed, one aim of this study is to demonstrate the ubiquitous characteristics of TME components among different cancer types or tissue types. We observed that the same major lineages of TME from different cancer types were clustered together according to the well-established similarity analysis, suggesting that the analysis on TME at pan-cancer level is actionable. To avoid the possibility of misleading, we also conducted analysis separately in each cancer type or divide the trajectories in terms of tissue derivation. As a result, we observed shared distribution of TME cells compared with biased distribution of cancer cells. On the other hand, confocal images of mIF are also provided in our revised version of manuscript. Please check our response one-by-one below.

Major comment

1. Page 7, lines 177-180: According to the original paper, early stage tumors are from surgical resection and late stage tumors from biopsies; many from lymph node biopsies and very few fibroblasts were recovered. Brain metastasis samples are resected from the brain. Due to the differences in the sampling sites and methods, the statement is over-interpretation.

Response: We thank the reviewer’s valuable comment. We totally agree that differences in the

sampling sites and method may bias the proportion of stroma components. In order to confirm specific trend of $FABP4^+$ macrophages among different sampling sites of lung cancer, we conducted the same analysis in other clusters of macrophages, which is illustrated below and in **Supplementary Fig 1e**. In detail, the macrophages were divided into 4 subclusters (i.e., $IL1B^+$, $SPP1^+$, $APOE^+$ and $FABP4^+$) according to its unique expression pattern. No similar trend was found in other three clusters of macrophages as that was observed in $FABP4^+$ macrophages (**Fig. 1g and Supplementary Fig. 1e**). Intriguingly, $SPP1^+$ macrophage was significantly higher in brain metastasized tissue than three other groups with similar proportions, which is consistent with the poor prognostic value of $SPP1$ in all types of cancer based on TCGA projects (see below). We revised our manuscript and modified the expression accordingly to avoid overinterpretation (line 181-185).

2. Page 8, lines 202-207 related to Supplementary Fig 1g: A single picture of co-localization is not very convincing. What proportion of CD11c+CD86+ populations co-express CD3? Providing flow cytometry data or confocal images will be better to exclude adjacent cells of CD11c+CD86+ and CD3+.

Response: We appreciated this suggestion. Due to the unavailability of fresh samples, we conducted mIF and get a series of confocal images to estimate the proportion of CD3⁺CD11c⁺CD86⁺ (CD3⁺ DC) in three cancer types (i.e., ATC, STAD and COAD). After panoramic quantifying a series of staining confocal images for FFPE slides from 9 patients (3 for each of anaplastic thyroid cancer, colorectal cancer and stomach cancer), proportion of CD3⁺ DC cells ranged from 3% to 13% of myeloid cell (CD11c⁺) (**Supplementary Fig. 1j and below**). Representative confocal images for each cancer type were provided to exclude the potential possibility of adjacent cells (**Supplementary Fig. 1i and below**). In the supplementary figure, the CD3⁺ T cell was indicated by blue arrow, while the CD3⁺DC (CD3⁺CD11c⁺CD86⁺) was indicated by red arrow. Additionally, CD3⁺ expression has been observed on small proportion of DC cell as early in 1989 through flow cytometry (N Romani *et al.*, J Exp Med, 1989). Therefore, the pan-cancer analysis facilitated the identification of such small proportional cells in cancer samples by pooling numerous cells at pan-cancer level, which highlights the ubiquity of such cell type among different cancer types. Collectively, our result supported the ubiquitous existence of CD3⁺ DC, which is worthwhile to explore in the further cancer study. We revised this part of our manuscript in line 210-216.

3. Lines 208-213 related to supplementary figure S1h: Cluster5 rarely express B cell markers and looks like mis-classified cells. Please provide cluster-level differentially expressed genes (DEGs) and check whether they are misclassified T cells or potential doublets.

Response: We thank the reviewer for this comment. First, cells in subcluster5 expressed the cluster-level DEG of cluster 13, which are B cell markers (e.g., Top DEGs: *CD79A*, *MS4A1*) provided in **Supplementary Table 2**. Feature plot of the marker genes were illustrated below. *CD79A* and *MS4A1* (encoding CD20) were also expressed in part of cells in subcluster5, which was the specific cluster-level DEG of cluster 13 (**Supplementary Table 2**) (see below).

Expression distribution of top marker genes in B lymphocyte cluster

Secondly, although the proportion of *MS4A1*⁺ cells were relatively low in subcluster5 (25.8%) than other B cell subclusters, it is much higher than the proportion of *MS4A1*⁺ in all T cells (2.2%), ranging 0.2% to 6.5% (see below left) of all T subclusters that described in Supplementary Fig. 4a. In addition, 20.02% of B cell subcluster5 was *MS4A1*⁺*CD3D*⁺ double positive, compared with 1.2% in all T cells (ranging from 0% to 2.1% of T cell subclusters) (see below right). Thus, the *CD3* and *MS4A1* are not absolutely exclusive expressed in T and B cells.

tt	pos		tt	pos	
	CD20 neg	CD20 pos		CD20,CD3D double pos	
CD8+ TEM	0.972450788	0.027549212	CD8+ TEM	0.0212355212	
CD4+ TCM	0.984557746	0.015442254	CD4+ TCM	0.0126079166	
Treg	0.992293775	0.007706225	Treg	0.0066340548	
Cycling CD4+ T	0.981834293	0.018165707	Cycling CD4+ T	0.0062029242	
DN T	0.935257758	0.064742242	DN T	0.0093672338	
CD8+ CTL	0.990691226	0.009308774	CD8+ CTL	0.0067340067	
PDCD1+ CD8+ T	0.990010447	0.009989553	PDCD1+ CD8+ T	0.0092060590	
Cycling CD8+ T	0.990125434	0.009874566	Cycling CD8+ T	0.0080064051	
Cycling NKC	0.988326848	0.011673152	Cycling NKC	0.0000000000	
TIM3+ NKC	0.998040070	0.001959930	TIM3+ NKC	0.0006533101	
CD160+ NKC	0.984539705	0.015460295	CD160+ NKC	0.0026704146	

Proportion of MS4A1+ cells in all T cell clusters

Interestingly, lymphocytes were automatically grouped into 9 clusters with unsupervised clustering, with only one was annotated as B cell (Fig. 1c and 1e). We extracted all the lymphocytes to illustrate the location of subcluster5 of B cells and noticed that this subcluster located between T cells and other B cells (see below). Due to the substantial difference of *MS4A1*⁺ proportion between subcluster5 of B cells and T cells, it may be automatically clustered into the sole B cell cluster.

Distribution of subcluster 5 of B cells and other B cells/T cells

Thirdly, we have excluded the potential doublets predicted by Scrublet (line 568-569) in the QC step. To excluded more possible doublets, we also conducted another software (i.e., Doubletfinder) to screen the potential doublets, identified 71 more potential doublet cells out of 1413 cells, covering 5.02% of subcluster5 of B cells. That means still 94.98% cells in subcluster5 were not doublet according to the current bioinformatic analysis approach. Indeed, CD3⁺CD20⁺ (i.e., CD3D⁺MS4A1⁺) lymphocyte cells have been experimentally identified as a small cell population through flow cytometry, but paradoxically referred as T cells or B cells due to its similarity to both cell types (Schuh E *et al*, *J Immunol*. 2016; Nagel A *et al*, *Plos One*. 2014). In Nagel A's report, they demonstrated CD3⁺CD20⁺ B cell as storage effect-antigen exchange between T and B lymphocyte. Thus, CD3 is not produced endogenously, but exchanged from T cell surface antigen when contact.

And in Schuh E's report, they named the small cell population as CD3⁺CD20⁺ T cell, which can produce different cytokines from other T cells.

Based on the evidence above, we considered the real existence of CD3D⁺MS4A1⁺ cells. However, due to the possibility of controversies through automatically unsupervised clustering approach, and our main focus is CAFs in this study, we reworded the description in our manuscript and just briefly illustrated the heterogeneity of B cells in line 215-216.

4. Figure 2a and b, Figure 3: Integrated pan-cancer analysis for cellular interaction is odd and misleading.

Response: We appreciate this comment. Indeed, one aim of this study is to demonstrate the ubiquitous characteristics of CAFs among different cancer types or tissue types. And TME cells are more homogenous among different individual samples than malignant epithelial cells. As a good example, Wu S et al. illustrated the single cell profile of breast cancer and claimed that "UMAP visualization showed a clear separation of epithelial cells by tumor...In contrast, UMAP visualization of stromal and immune cells across tumors clustered together without batch correction" (Wu S et al, *Nat Genet*, 2021) (see below)

Figure of "Wu S et al, *Nat Genet*, 2021", epithelial cells from each breast cancer patient were separated by different color.

More importantly, pan-cancer single cell transcriptome atlas of tumor infiltrating T cells and myeloid cells have been illustrated recently (Cheng S et al. *Cell*, 2021; Zheng L et al. *Science*. 2021), claiming "As expected, the same major lineages from different cancer types were clustered together, further demonstrating that major myeloid lineages shared similar transcriptomic profiles". Therefore, we followed the same pipeline in the Cell paper, and conducted the similarity analysis to confirm the similarity of all TME components as well as CAFs clusters among different cancer types (Supplementary Fig. 2a and 3a, also see below).

Similarity of different TME components among diverse cancer types

Similarity of different CAF clusters among diverse cancer types

Moreover, to exclude the possibility of misleading and the possible dominant effects of normal/adjacent samples from 4 cancer types, we also conducted cellular interaction analysis in each cancer type separately (see below). It is consistent and representative that the overall interaction increased in the order of normal, adjacent, and tumor tissue.

Interaction counts of different TME components among normal, adjacent and tumor samples separately in terms of cancer types

Also, fibroblasts presented the most prolific interactions with other components in tumor/adjacent samples but not normal samples regardless of the tissue type (**Supplementary Fig. 2b** and below). In this figure, increased interaction from each type of tissues (normal, adjacent and tumor) can also be discriminated according to the maximal interaction counts of y axis.

Interaction comparison in **tumor** tissue

Interaction comparison in **adjacent** tissue

Interaction comparison in **normal** tissue

Tissue type-specific interaction quantification of the main TME components represented in tumor, adjacent, and normal tissues

Finally, we reword our manuscript to avoid misleading or overinterpretation. We highlighted our aim of investigation on ubiquitous characteristics of TME in line 146-147, line 170-172, line 223-226, demonstrated the similarity of CAFs clusters among cancer types in line 237-241, and the most prolific interactions of fibroblasts with other TME cells in line 232-236.

5. Figure 2f-h, Figure 4d: Trajectory may be used the gene expression comparisons shared between different cancer types. However, integrated pan-cancer trajectory analysis is odd and misleading for the inference of cellular transformation.

Response: We appreciate this comment. Similar to Q4 described above, one aim of this study is to demonstrate the possible ubiquitous evolution trajectories of TME, especially CAFs among

different cancer types. As described above, we followed the same pipeline of previous report (Cheng S *et al. Cell*, 2021. Zheng L *et al. Science*, 2021), and conducted the similarity analysis to confirm the similarity of CAFs clusters among different cancer types (**Supplementary Fig. 3a** and below). We also conduct the similarity analysis for TME components, including macrophages (below), which is consistent with previous report (Cheng S *et al. Cell*, 2021). Therefore, we speculated that the similarity of TME components was the basis of pan-cancer trajectory.

Similarity analysis of CAF clusters among different cancer types

Similarity analysis of myeloid cells clusters among different cancer types

Therefore, we performed integrated pan-cancer trajectory analysis because these clusters were shared with all cancer types, and exhibited the similar marker genes. To better clarify the generality of CAFs states, we added the constitution ratio of cancer type in each state as **Supplementary Fig.3b**. All cancer types can be found in each CAFs state (**Supplementary Fig. 3b** and below), except for the unavailability of normal breast tissue profile.

The constitution ratio of each cancer type in each state

Furthermore, to avoid the possible misleading, we also divided the trajectories in terms of tissue derivation, observing the shared distribution of cell stated along the trajectories in different cancer types (below). Therefore, we considered that the CAFs evolution was shared among cancer types. We reword our manuscript to avoid misleading or overinterpretation. We highlighted our aim of

investigation on ubiquitous characteristics of TME in line 146-147, line 170-172, demonstrated the similarity of CAFs clusters among cancer types in line 237-241, and the cancer type constitutions in each CAF state in line 270-271.

Cell distribution of three CAF states along evolutionary trajectory in different cancer types

6. Figure 4d: A single picture of co-localization is not very convincing. What proportion of α -SMA+ populations co-express CD163? Providing flow cytometry data or number of confocal images will be more convincing.

Response: We thank the reviewer's great suggestion. Due to the lack of available fresh tumor samples, we performed mIF experiments and took confocal image to estimate the proportion of α -SMA⁺ with co-expressed CD163 in three cancer types (anaplastic thyroid cancer, colorectal cancer, and gastric cancer) with available FFPE slides. Totally 6 FFPE slides of tumor samples from 6 patients were co-stained with α -SMA and CD163. After evaluating 66 confocal images from these 6 slides (10 to 12 images per slides), the proportion of double positive cells (refer to CAF_{ap}) ranged from 10.2% to 13.6% normalized to all α -SMA⁺ cells in three cancer types. The represented confocal images were provided in **Fig. 4d** and **Supplementary Fig. 5c** and below.

Confocal image of α -SMA and CD163 staining in three cancer types

The result was consistent with the previous result in pancreatic cancer (Elyada E *et al. Cancer Discov.* 2019. See below). Intriguingly, a latest study (published on Apr. 26 2022 after submission of our manuscript) also demonstrated" the existence of CAF_{ap} (named as apCAF in this study), and its regulatory role on Treg in pancreatic cancer through experimental approaches (Huang H *et al. Cancer Cell*, 2022). Therefore, we illustrated the ubiquitous of CAF_{ap} among all types of cancer through pan-cancer analysis. We revised our manuscript in line 364-375.

[Redacted]

Figure of “Elyada E *et al. Cancer Discov. 2019*” Showed the percentage of CAF_{ap} in all CAFs.

7. Figure 5f: Is it possible to distinguish between signatures of EndMT and of abundant CAFs and endothelial cells? If possible, please provide the unique EndMT signature not detected in CAFs or endothelial cells.

Response: We appreciated this suggestion. We performed DEG analysis and venn diagram to detect the unique signature. The unique signature of CAF_{EndMT} were listed in **Supplementary table 7** and the heatmap was listed in **Supplementary Fig. 6b** and below. We also presented the violin plot of *ESM1*, which is an example of unique marker of CAF_{EndMT} and has been already reported as an angiogenesis-related gene in endothelial tip cells (Rocha S *et al. Circ Res. 2014*). Additionally, we also added the GO enrichment in the **Supplementary Fig. 6c**, which indicated that the enrichment of CAF_{EndMT} in ameboidal-type cell/epithelial cell migration etc. The manuscript has been revised accordingly in line 401-405.

Heatmap of differential expressed genes in CAF_{EndMT} and the representative marker

The KEGG pathway enrichment plot of CAF_{EndMT}

8. Line 434-436 related to Figure 6h: the resolution of spatial transcriptomics is ~10 cells and does not allow distinction of co-localization of two cell types from a single cell type with co-expression of the genes. It should be clearly noted in the text to avoid over-estimation of the data.

Response: We thank this reviewer's valuable comment. We totally agree that co-localization of two cell types and a single cell type with co-expression of the genes would result in the same pattern because the resolution of spatial transcriptomics is about 10 cells. Therefore, to exclude the interference of co-expression, we calculated the signature of CAF_{EndMT} and SPP1⁺TAM with spatial transcriptomic data based on AddModuleScore algorithm. Consistently, significantly positive correlation was observed between CAF_{EndMT} score and SPP1⁺TAM score ($R=0.21$, $p < 2.2 \times 10^{-16}$), suggesting the co-localization of these two cell types. Therefore, we revised the Fig.6h (below) and the description in the manuscript in line 461-466.

Correlation of CAF_{EndMT} and SPP1⁺TAM signatures according to spatial transcriptome

Reviewer #2 (Remarks to the Author): Expert in single-cell RNA-seq and computational cancer genomics

The manuscript presents a pan-cancer single cell analysis of the tumor microenvironment, with a focus on fibroblasts. The authors have a great grasp of vexing issues around fibroblasts. They provided an excellent framing of these issues in the introduction, particularly on topics of fibroblast plasticity and interactions in TME.

The main strength of this study is the work to harmonize multiple scRNAseq data sets across different cancer and provide pancancer clusters. This harmonized data will be of interest to the broader community and likely regarded as an important resource.

The authors use the harmonized scRNAseq data for exploration and discovery with a focus on the nonmalignant compartment of the tumor microenvironment (TME). The analyses are interesting but mostly descriptive (i.e. proportion of cell types under different conditions, expression of canonical markers), and lack statistical rigor. The authors have the sample size to provide statistical significance but few p-values are reported. Adding statistical significance to all claims is needed for this study.

Response: We thank the reviewer's valuable comments. We have conducted the statistical analysis as possible as we can, and revised our manuscript accordingly. We conducted statistical analysis and illustrated the p value for Fig. 1g, Fig. 1h, Fig. 1j, Fig. 2a, Fig. 2c, Fig. 2g, Fig. 3g, Fig. 4b, Fig. 4g, Fig. 5b, Fig. 5f, Fig. 6f, Fig. 6g, Fig. 6h and several supplementary Figs.

The authors make at least three scientific claims that are backed up to varying degrees. First, the authors claim to identify new cell types, namely CD3+ DC and Bregs. Claiming new cell types from scRNAseq data is very tricky business. Limited IF images are provided for the CD3+ DC and nothing for the Bregs. I recommend these speculative components are moved to the supplement.

Response: We appreciate this comment and suggestion and totally agree that claiming new cell types from scRNA-seq data is tricky. Because our main focus is CAFs among different cancer types, so we moved the descriptions on other cell types into supplementary figures and reword the description of the possible existence of the new cell types to avoid overinterpretation. Indeed, the "new" cell types described in our manuscript have been reported previously through flow cytometry, including CD3+ DC cells, CD3+CD20+ cells etc (Romani N *et al.*, *J Exp Med*, 1989; Schuh E *et al*, *J Immunol*, 2016). We also updated the citation accordingly.

On the other hand, we provided more evidence to support the possible existence of the small cell clusters as the requirement of another reviewer through evaluating a series of confocal images for mIF. In detail, to better illustrate CD3+DC, we provided the confocal images in **Supplementary Fig. 1i** (also see below). The CD3+T cell was indicated by blue arrow, while the CD3+DC (CD3+CD11c+CD86+) was indicated by red arrow.

Additionally, as early as in 1989, CD3+DC was reported (Romani N *et al. J Exp Med*, 1989), and CD3 was reported expressed on small proportion of DC cells. Staining quantification results, normalized by the number of myeloid cell (CD11c+), the proportion of CD3+CD11c+CD86+ (CD3+DC) cells ranged from 3% to 13% (**Supplementary Fig.1j** and below). Therefore, the pan-cancer analysis facilitated the identification of such small proportional cells by pooling numerous cells from different cancer types. Finally, we reword the description of the existence of CD3+DC to avoid overinterpretation in line 210-216.

As the CD3⁺ B cell.

First, cells in subcluster5 expressed the cluster-level DEG of cluster 13, which are B cell markers (e.g., Top DEGs: *CD79A*, *MS4A1*) provided in Supplementary Table 2. Feature plot of the marker genes were illustrated below. *CD79A* and *MS4A1* (encoding CD20) were also expressed in part of cells in subcluster5, which was the specific cluster-level DEG of cluster 13 (Supplementary Table 2) (see below).

Expression distribution of top marker genes in B lymphocyte cluster

Secondly, although the proportion of *MS4A1*⁺ cells were relatively low in subcluster5 (25.8%) than other B cell subclusters, it is much higher than the proportion of *MS4A1*⁺ in all T cells (2.2%), ranging 0.2% to 6.5% (see below left) of all T subclusters that described in Supplementary Fig. 4a. In addition, 20.02% of B cell subcluster5 was *MS4A1*⁺*CD3D*⁺ double positive, compared with 1.2% in all T cells (ranging from 0% to 2.1% of T cell subclusters) (see below right). Thus, the *CD3* and *MS4A1* are not absolutely exclusive expressed in T and B cells.

tt	pos		tt	pos	
	CD20 neg	CD20 pos		CD20,CD3D double pos	
CD8+ TEM	0.972450788	0.027549212	CD8+ TEM	0.0212355212	
CD4+ TCM	0.984557746	0.015442254	CD4+ TCM	0.0126079166	
Treg	0.992293775	0.007706225	Treg	0.0066340548	
Cycling CD4+ T	0.981834293	0.018165707	Cycling CD4+ T	0.0062029242	
DN T	0.935257758	0.064742242	DN T	0.0093672338	
CD8+ CTL	0.990691226	0.009308774	CD8+ CTL	0.0067340067	
PDCD1+ CD8+ T	0.990010447	0.009989553	PDCD1+ CD8+ T	0.0092060590	
Cycling CD8+ T	0.990125434	0.009874566	Cycling CD8+ T	0.0080064051	
Cycling NKC	0.988326848	0.011673152	Cycling NKC	0.0000000000	
TIM3+ NKC	0.998040070	0.001959930	TIM3+ NKC	0.0006533101	
CD160+ NKC	0.984539705	0.015460295	CD160+ NKC	0.0026704146	

Proportion of MS4A1+ cells in all T cell clusters

Interestingly, Lymphocytes were automatically grouped into 9 clusters with unsupervised clustering, with only one was annotated as B cell (Fig. 1c and 1e). We extracted all the lymphocytes to illustrate the location of subcluster5 of B cells and noticed that this subcluster located between T cells and other B cells (see below). Due to the substantial difference of MS4A1+ proportion between subcluster5 of B cells and T cells, it may be automatically clustered into the sole B cell cluster.

Distribution of subcluster 5 of B cells and other B cells/T cells

Thirdly, we have excluded the potential doublets predicted by Scrublet (line 568-569) in the QC step. To excluded more possible doublets, we also conducted another software (i.e., Doubletfinder) to screen the potential doublets, identified 71 more potential doublet cells out of 1413 cells, covering 5.02% of subcluster5 B cells. That means still 94.98% cells in subcluster5 B cell were not doublet according to the current bioinformatic analysis approach. Indeed, CD3⁺CD20⁺ (i.e., CD3D⁺MS4A1⁺) lymphocyte cells have been experimentally identified as a small cell population through flow cytometry, but paradoxically referred as T cells or B cells due to its similarity to both cell types (Schuh E *et al*, *J Immunol*. 2016; Nagel A *et al*, *Plos One*. 2014). In Nagel A's report, they demonstrated CD3⁺CD20⁺ B cell as storage effect-antigen exchange between T and B lymphocyte.

Thus, CD3 is not produced endogenously, but exchanged from T cell surface antigen when contact. And in Schuh E's report, they named the small cell population as CD3⁺CD20⁺ T cell, which can produce different cytokines from other T cells.

Based on the evidence above, we considered the real existence of *CD3D⁺MS4A1⁺* cells. However, due to the possibility of controversies through automatically unsupervised clustering approach, and our main focus is CAFs in this study, we reworded the description in our manuscript and just briefly illustrated the heterogeneity of B cells in line 215-216.

Second, the authors claim that they have identified a fibroblast cell state that is associated with immunotherapy response. I found this part of the work most compelling, particularly the survival association. For a more convincing result, the author need to determine if the fibroblast cell type of interest added prognostic significance independent of other covariates. If so, I recommend that the authors provide richer insights (likely cell-cell interactions?) to support this association.

Response: We appreciated this valuable suggestion and performed multi-variate analysis. We have added it in the **Supplementary Fig. 4e** (also see below), CAF_{state3} enrichment score exhibited its independent prognostic value on outcome of PD-1 treatment after adjustment with available clinical information in each cohort. We revised the manuscript accordingly in line 338-341.

Additionally, the cell-cell interactions between CAF states and TME components (NK, T, DC, and macrophages) has been shown in **Fig. 3** and **Supplementary Fig.4**. In particular, a series of interactions between LGALS9 (encoding galectin-9) and its partners are enriched in cross talk between T cell/macrophage/endothelial and CAF_{state3} but not CAF_{state1}/CAF_{state2}. Interestingly, galectin-9 related interactions have been linked to tolerogenic macrophage programming and adaptive immune suppression (Daley D *et al*, *Nat Med*. 2017), and regulate T cell death for cancer immunotherapy (Yang R *et al*, *Nat Commun*. 2021). These interactions may partially support the specific association of CAF_{state3} with immunotherapy response. We reworded the manuscript accordingly in line 325-331 and cited the reference above.

Urothelial carcinoma

Uterine Sarcomas

Melanoma

Multivariate analysis of factors associated with survival outcomes of immunotherapy cohorts

Third, the authors consider CAF transdifferentiation through ENDOMT and the TAM plasticity. This work is intriguing but not presented in a statistically rigorous manner. Here the authors propose several hypotheses but do not frame their analysis as a hypothesis test to make this aspect of their work convincing. One idea to consider creating null distributions from all the cell types not likely to be associated with transdifferentiation and derive a p-value for the findings reported.

Response: We appreciate this valuable comment. As the reviewer's suggestion, we established null control with other cell types. We used TAM (c19), CAF_{myo} (c1) and all other clusters (except the CAF_{ap}) to explore the possibility of transdifferentiation. We illustrated all the trajectories below. Distinct two paths were exhibited compared to the gradual trajectory of TAM to CAF_{myo} via CAF_{ap}. However, we can't conduct statistical analysis with this approach.

Alternatively, we conducted permutation analysis to estimate the significance of the transdifferentiation. The PC1 value of cells in TAM-CAF_{ap}-CAF_{myo} path and TEC-CAF_{EndMT}-CAF_{myo} path is illustrated below, exhibiting significance with linear regression model. We thus shuffled the annotation of each cell and conducted linear regression analysis for 1,000,000 times, less than 5% of statistical results have $p < 0.05$, and none has p value less than the indicated p value (i.e., $p < 1.4 \times 10^{-137}$ for TAM-CAF_{ap}-CAF_{myo} and $p < 1 \times 10^{-202}$ for TEC-CAF_{EndMT}-CAF_{myo} shown below). Therefore, with this permutation test, we confirmed the significance of transdifferentiation path.

Linear regression analysis of PC1 value of two transdifferentiation paths

Throughout the entire manuscript claims of casuality are substantially overstated. There is no mechanistic work in this study so the authors can not make claim that, for example, a specific subset of fibroblasts “sculpts” the TME. Claims of casuality from presumed cell-cell interactions and pseudo time trajectories are misleading without any mechanistic work.

Response: We appreciate this comment and totally agree with that there’s no mechanistic work in this study. Therefore, we have reworded our descriptions all through the manuscript and highlighted the importance of future mechanistic work to establish the causality to avoid substantial overstatement (e.g., line 470, line 535-536). Particularly, we added description on limitation of our study on mechanistic work in the discussion part (line 520-529)

A major missing component of this work is the interaction of non-malignant cells with malignant cells in the TME. There is surprisingly no consideration tumor-fibroblast interactions. Moreover, the authors considerations of cell-cell interactions in a pancancer context is limited. There is little exploration findings that are shared or not shared across different cancer types. Exploration of fibroblast-malignant interactions would strengthen the study, especially if reported in a pancancer context.

Response: We appreciated this suggestion. We also considered the same study designing problem at first. However, we did such design with the following reasons.

First, our main goal of this study is to demonstrate the ubiquitous characteristics among different cancer types or tissue types, but not the difference between cancer types and samples. We believe the exploration commonalities, but not the difference at pan-cancer level will be more important to facilitate pan-cancer treatment, like immunotherapy.

Secondly, we totally agree that exploration of fibroblast-malignant interactions is very important to establish the important role of CAFs among different cancer types. However, both previous reports and our study have revealed the ubiquitous characteristics of TME components, but not the malignant cells. For instance, previous report on single-cell profile of breast cancer, claiming that “a clear separation of epithelial cells by samples. In contrast, UMAP visualization of stromal and immune cells across tumors clustered together without batch correction” (Wu S *et al*, *Nat Genet*, 2021) (see the original figure below), suggesting the huge heterogeneity of tumor cells even in the same cancer type. So does in our pan-cancer study. The epithelial cells, even within the same type, had an obvious biased distribution.

Figure of “S Wu et al, Nat Genet, 2021”, epithelial cells from each breast cancer patient were separated by different color

In our pan-cancer study, we also observed obvious biased distribution of malignant epithelial cells compared to the ubiquitous distribution of TME cells (**Supplementary Fig. 1c and 1d**, also see below), indicating it is better to explore fibroblast-malignant interactions in specific cancer type separately but not at pan-cancer level, which has already been revealed individually for each cancer type in the original data derived papers. We added the description of bias distribution of epithelial cancer cells in line 170-173

Epithelial cells distribution among different cancer types (**Supplementary Fig. 1d**)

TME cells distribution among different cancer types (**Supplementary Fig. 1c**).

Moreover, the ubiquitous characteristics of TME clusters among different cancer types according to the similarity analysis, suggested that it is actionable to conduct analysis at pan-cancer level (**Supplementary Fig. 2a** and below). Therefore, we only illustrate the ubiquitous characteristics and interactions between different TME components but not interactions of TME cells with malignant cells due to the diverse characteristics of malignant cells among different cancer types.

Similarity analysis of TME clusters among different cancer types

Reviewer #3 (Remarks to the Author): Expert in cancer-associated fibroblasts, tumour microenvironment, and cell plasticity

In this manuscript, Luo et al. analyze normal fibroblasts and CAFs across multiple tissue/cancer types. Three distinct differentiated states of CAFs were identified with distinct associations with response to immunotherapy. In addition, the authors propose that CAFs arise from macrophages, peripheral nerve, and endothelial cells. While this manuscript provides a comprehensive analysis of CAFs across cancer types, the evidence for the origins of CAFs and their presumed functions is lacking. Several points should be addressed, as outlined below.

1. The authors identify a subset of regulatory B cells that express CD3D, IL32, and TIGIT (cluster 5); however, this cluster expresses low levels of CD79A and MS4A1, suggesting that this subset may be misclassified as B cells. Expression of CD79A and MS4A1 in T cell subsets should be included to support the argument that this cluster is in fact B cells.

Response: We thank the reviewer for this comment. First, cells in subcluster5 expressed the cluster-level DEG of cluster 13, which are B cell markers (e.g., Top DEGs: *CD79A*, *MS4A1*) provided in Supplementary Table 2. Feature plot of the marker genes were illustrated below. *CD79A* and *MS4A1* (encoding CD20) were also expressed in part of cells in subcluster5, which was the specific cluster-level DEG of cluster 13 (**Supplementary Table 2**) (see below).

Expression distribution of top marker genes in B lymphocyte cluster

Secondly, although the proportion of *MS4A1*⁺ cells were relatively low in subcluster5 (25.8%) than other B cell subclusters, it is much higher than the proportion of *MS4A1*⁺ in all T cells (2.2%), ranging 0.2% to 6.5% (see below left) of all T subclusters that described in **Supplementary Fig. 4a**. In addition, 20.02% of B cell subcluster5 was *MS4A1*⁺*CD3D*⁺ double positive, compared with 1.2% in all T cells (ranging from 0% to 2.1% of T cell subclusters) (see below right). Thus, the *CD3* and *MS4A1* are not absolutely exclusive expressed in T and B cells.

tt	pos		tt	pos	
	CD20 neg	CD20 pos		CD20,CD3D double pos	
CD8+ TEM	0.972450788	0.027549212	CD8+ TEM	0.0212355212	
CD4+ TCM	0.984557746	0.015442254	CD4+ TCM	0.0126079166	
Treg	0.992293775	0.007706225	Treg	0.0066340548	
Cycling CD4+ T	0.981834293	0.018165707	Cycling CD4+ T	0.0062029242	
DN T	0.935257758	0.064742242	DN T	0.0093672338	
CD8+ CTL	0.990691226	0.009308774	CD8+ CTL	0.0067340067	
PDCD1+ CD8+ T	0.990010447	0.009989553	PDCD1+ CD8+ T	0.0092060590	
Cycling CD8+ T	0.990125434	0.009874566	Cycling CD8+ T	0.0080064051	
Cycling NKC	0.988326848	0.011673152	Cycling NKC	0.0000000000	
TIM3+ NKC	0.998040070	0.001959930	TIM3+ NKC	0.0006533101	
CD160+ NKC	0.984539705	0.015460295	CD160+ NKC	0.0026704146	

Proportion of *MS4A1*⁺ cells in all T cell clusters

Interestingly, Lymphocytes were automatically grouped into 9 clusters with unsupervised clustering, with only one was annotated as B cell (**Fig. 1c** and **1e**). We extracted all the lymphocytes to illustrate the location of subcluster5 of B cells and noticed that this subcluster located between T cells and other B cells (see below). Due to the substantial difference of *MS4A1*⁺ proportion between subcluster5 of B cells and T cells, it may be automatically clustered into the sole B cell cluster.

Distribution of subcluster 5 of B cells and other B cells/T cells

Thirdly, we have excluded the potential doublets predicted by Scrublet (line 568-569) in the QC step. To excluded more possible doublets, we also conducted another software (i.e., Doubletfinder) to screen the potential doublets, identified 71 more potential doublet cells out of 1413 cells, covering 5.02% of subcluster5 B cells. That means still 94.98% cells in subcluster5 B cell were not doublet according to the current bioinformatic analysis approach. Indeed, $CD3^+CD20^+$ (i.e., $CD3D^+MS4A1^+$) lymphocyte cells have been experimentally identified as a small cell population through flow cytometry, but paradoxically referred as T cells or B cells due to its similarity to both cell types (Schuh E *et al*, *J Immunol*. 2016; Nagel A *et al*, *Plos One*. 2014). In Nagel A's report, they demonstrated $CD3^+CD20^+$ B cell as storage effect-antigen exchange between T and B lymphocyte. Thus, CD3 is not produced endogenously, but exchanged from T cell surface antigen when contact. And in Schuh E's report, they named the small cell population as $CD3^+CD20^+$ T cell, which can produce different cytokines from other T cells.

Based on the evidence above, we considered the real existence of $CD3D^+MS4A1^+$ cells. However, due to the possibility of controversies through automatically unsupervised clustering approach, and our main focus is CAFs in this study, we reworded the description in our manuscript and just briefly illustrated the heterogeneity of B cells in line 215-216.

2. The expression of SPP1 and C1QC shown in in Figure 1 does not support the subclustering of SPP1+ TAMs and C1QC TAMs, due to the extensive overlap and poor segregation in the current setting. Also, all the other TAM markers were not shown.

Response: We appreciate this comment and sorry for the confusing of our original description. Actually, we did not perform the subclustering analysis in this part. $SPP1^+$ and $C1QC^+$ TAM was firstly mentioned in previous pan-cancer myeloid cell study, defining SPP1+ TAM as pro-angiogenetic TAM (SJ Cheng *et al*. *Cell*. 2021). However, they did not include any bench validation.

Therefore, we further explained the potential pro-angiogenic mechanism in the analysis- SPP1⁺ TAM promoted angiogenesis maybe through promoting endothelial-mesenchymal transition. We totally agree that some SPP1⁺ cells and C1QC⁺ cells are overlapped, that's why we did not do sub-clustering in our study but only compared the distribution of SPP1⁺/C1QC⁻ and SPP1⁻/C1QC⁺ cells in normal vs. tumor. To avoid the possibility of misleading, we provided the analysis for four cell groups (SPP1⁺/C1QC⁺, SPP1⁻/C1QC⁻, SPP1⁺/C1QC⁻, and SPP1⁻/C1QC⁺) (Fig. 1i and below) and modified our description in the main text (line 192-195). On the other hand, we have provided the markers for all clusters in **Supplementary Table 2**, including TAM (c19).

3. The authors state that the distribution of fibroblast clusters is altered in tumor and adjacent tissue compared to normal tissue, but it is difficult to appreciate the relative distribution of fibroblasts based on the way the data is currently presented. Including the clustering of fibroblasts in normal, tumor, and adjacent tissues would clarify differences in the tissue types and also how normal fibroblasts were defined (i.e., if normal fibroblasts are a largely distinct subset from adjacent and tumor-associated fibroblasts). Specifically, in Figure 2, what are the defining (unique) marker genes of NFs, in contrast to CAFs?

Response: We thank the reviewer for this comment. The distribution information of fibroblasts in normal, adjacent and tumor tissue mainly comes from Fig. 1e (see below left), which is the normalized proportion of sample derivation in each cluster. On the other hand, the tissue type-based distribution of fibroblasts was according to Fig.1d (see below right). To better appreciation of the distribution, we provided the relative normalized distribution percentage of cells among different tissue types and malignant states in Fig. 1d and **Supplementary table 3**.

As the consensus of CAFs (Sahai E *et al. Nat Rev Cancer, 2020*), it is difficult in defining fibroblasts results largely from the lack of unique markers that are not expressed in any other cell types. The result is that in practical terms, fibroblasts are often defined by a combination of their morphology, tissue position and lack of lineage markers for epithelial cells, endothelial cells, and leukocytes. Due to the lack of a unique panel maker to distinguish normal fibroblast and CAFs, we defined normal fibroblasts mainly according to the tissue-origin. As in Fig. 1e (see below left), it is clear that cells from normal samples are dominate in c3 and c5 (refer to NF), while cells from

tumor/adjacent samples are dominant in other clusters (refer to CAF). We reworded our description in line 237-241. Therefore, we compared the empirical maker of NFs and CAFs in **Fig.2c** and presented in **Supplementary Fig. 2c** and **Supplementary table 4**.

4. The similarities and differences of CAFs (and CAF subpopulations) across various cancer types appear to be overlooked. Is the CAF-state (state 1, state 2, state 3) universal to various cancer types?

Response: We appreciate this comment. We performed integrated pan-cancer trajectory analysis because these clusters were shared with all cancer types. For CAF states, all cancer types can be found in each state (**Supplementary Fig.3b** and below). We added the description in line 270-271.

The constitution ratio of each cancer type in each state

To avoid the possible misleading, we also divided the trajectories in terms of tissue derivation, observing the shared distribution of cell stated along the trajectories in different cancer types (see below). Therefore, we considered that the three CAF states are universal to various cancer types. And to avoid the overstatement, we have reworded the according part in the manuscript.

Cell distribution of three CAF states along evolutionary trajectory in different cancer types

5. Some of the fibroblast clusters identified express high levels of ACTA2 and RGS5 (cluster 1 and CAFEndoMT) which the authors refer to as canonical fibroblast markers. While ACTA2 and RGS5 mark fibroblasts, they are also expressed in perivascular smooth muscle cells/pericytes and the cells detected in these subsets may not be fibroblasts derived from EndMT. Consequently, this caveat should be included in the manuscript or additional evidence that these subsets are not pericytes provided.

Response: We appreciate the reviewer for this valuable comment. We totally agree that ACTA2 and RGS5 are expressed in both fibroblast and perivascular smooth muscle cells/pericytes, partially because both of them origin from mesenchyme and exhibited shared expression of some genes. For instance, RGS5 expression was defined as one of the markers to identify pericytes in murine breast cancer (Bartoschek M *et al. Nat Commun. 2018*). Additionally, it is well established that up to 40% of CAFs might be derived through EndMT (Zeisberg E *et al. Cancer Res. 2007*), and pericytes are the important resource of CAFs in tumors (Hosaka K *et al. Proc Natl Acad Sci USA. 2016*). Pericyte recruitment by EndMT (Hosaka K *et al. Proc Natl Acad Sci USA. 2016*) or direct pericyte-fibroblasts transition (Choi S *et al. Nat Commun. 2018*) led to tumor vasculature. In all, pericytes are an important origin of CAFs depended on EndMT (mainly) or pericyte-fibroblast transition. Though demarcating the two process is not the main topic in our manuscript, which was the topic in previous study (Lars Muhl *et al. Nat Commun. 2020*), we mentioned this possibility and revised manuscript to avoid overstatement in discussion part in line 509-529.

6. PDGFRA, PDGFRB, FAP, NOTCH3, HES4, and THY1 were enriched in CAFs compared to NFs, which the authors attribute to differences in functions related to angiogenesis and immunomodulation, but the majority of these genes are CAF biomarkers (PDGFRA, PDGFRB, FAP, and THY1). Similarly, genes highly expressed in CAFstate2 (PDGFRB) and CAFstate3 (PDGFRA, ENG, and FAP) are CAF markers. As a result, increased expression is not necessarily related to functions in angiogenesis or immunomodulation, rather an indication of increased activation into CAFs. The conclusions about potential functions should be revised or additional non-CAF biomarker genes included to support this hypothesis.

Response: We appreciated this important comment. We totally agree that increased expression is not necessarily related to functions rather an indication of increased activation into CAFs. Due to the heterogeneity of CAFs (Sahai E *et al. Nat Rev Cancer. 2020*), specific high expression of some

specific genes has been noticed as canonical makers of CAF subclusters, and thus used to define these CAF subclustered, like FAP⁺CAF, THY1⁺CAF and ENG⁺CAF. Increasing experimental evidence has revealed the potential bio-functions of these CAFs subclusters in specific diseases or cancers. For example, ENG⁺ (CD105) fibroblasts were permissive for pancreatic tumor growth in vivo (Hutton C *et al. Cancer Cell. 2021*). THY1⁺ (CD90) fibroblasts play important inflammation regulation role in rheumatoid arthritis (Mizoguchi F *et al. Nat Commun. 2018*), FAP⁺THY1⁺ fibroblasts would promote persistent inflammation arthritis (Croft A *et al. Nature. 2019*), FAP⁺ fibroblasts can promote immunosuppression in multiple cancers (liver, lung and pancreatic cancers) (Kraman M *et al. Science. 2010*; Yang X *et al. Cancer Res. 2016*) and induced angiogenesis by affecting the balance of pro-angiogenic and anti-angiogenic mediators in lung cancer (Santos A *et al. J Clin Invest. 2009*), and PDGF signaling of CAFs may not only stimulate tumor angiogenesis but also mediate resistance to anti-angiogenic therapy (Crawford Y *et al. Cancer Cell. 2009*; Pietras K *et al. PLoS Med. 2008*). Additionally, the different clusters of CAFs also involved in various function by genes enrichment (**Supplementary Fig.2e**). Thus, specific marker of CAFs stands for some biofunctions of CAFs in some degree according to current available evidence.

In our study, we illustrated the ubiquitous existence of different CAFs subclusters among different cancer types. Given functions of these clusters have been revealed through previous mechanistic reports in specific cancer type, we thus suggested the possible shared functions of these clusters in different cancer types. It would improve classification of heterogeneous CAFs and facilitate the understanding of them. Since we didn't perform functional validation, we modified our expression, cited the important references, and highlighted the importance of future mechanistic work to avoid substantial overstatement (e.g., line 470, line 520-529).

7. In the same vein, the correlation between CAFs and TAMs is only descriptive, without any functional validation. This should be revised to avoid over interpretation of the results.

Response: We appreciate this comment and totally agree that we don't have any functional validation on CAFs and TAMs. Intriguingly, a latest study (published on Apr. 26 2022 after submission of our manuscript) also demonstrated the existence of CAF_{ap} (named as apCAF in this study), and its regulatory role on Treg in pancreatic cancer through functional validation (Huang H *et al. Cancer Cell, 2022*). Therefore, we illustrated the ubiquitous of CAF_{ap} among all types of cancer through pan-cancer analysis. We have reworded our description in the manuscript to avoid overinterpretation and illustrated the latest study in line 364-375. Also, we added description on limitation of our study in the discussion part (line 509-529)

8. In Figure 5, it is unclear why the EC and CAF clusters are closely adjacent, in contrast to all other studies showing clear, distinct segregation between these two clusters. This raises questions with respect to the data analysis methodology of this study. Where are EC and CAF clusters located in Figure 1C? Also, representative EC and CAF marker genes are missing here. The CAF-endMT conclusion is based on descriptive data in Figure 5, without any functional validation. This should be revised to avoid over interpretation of the results.

Response: We appreciate this comment. We're sorry that the location of EC and CAF is not very clearly illustrated in **Fig.1c** because a total of 34 clusters are labeled at the same time. That's why we combined all fibroblast clusters and endothelial clusters in the right side of **Fig.1c** (see below)

and specifically took out the CAFs and ECs to illustrate in **Figure 5a**. Please check the location of c1 (CAF_{myo}), c6 (CAF_{EndMT}) and c25 (TEC) below.

Relative location of c1, c6 and c25 in UMAP

Representative markers of metaclusters of endothelial and fibroblast cells have been provided in **Fig. 1c** (see below), and the makers genes for each cluster have also been provided in **Supplementary Table 2**.

Main clusters and representative markers

For data analysis methodology, the most popular software to reduce dimension is t-SNE and UMAP currently. According to the previous reports, distance reflects similarity between clusters in UMAP but not in t-SNE. Meanwhile, UMAP can represent the continuous cell trajectory better than t-SNE. (Becht E *et al. Nat Biotechnol. 2018*). Therefore, more and more studies use UMAP instead of t-SNE in recent single-cell studies. For ECs and fibroblasts, a series of UMAP-based studies consistently illustrated the short distance between fibroblast and endothelial cells (Li X *et al. Theranostics. 2022*; Davidson S *et al. Cell Rep. 2020*; Kumar V *et al. Cancer Discov. 2022*; Krishna C *et al. Cancer Cell. 2021*), compared to segregated location with t-SNE-based studies. Similarly, the CAFs and ECs also segregated if applied t-SNE clustering in our study (see below).

Figures from four UMAP-based studies indicated the short distance between endothelial and fibroblast

Comparison of t-SNE-based and UMAP-based clustering in our study

To better illustrate and avoid confusion, we have revised the table and the according part, and reworded our descriptions to avoid overinterpretation.

9. In addition, can the author clarify whether the correlation between CAF-endMT and SPP1 TAM

is applicable to other cancer types in clinical datasets such as TCGA?

Response: We thank the reviewer for this great suggestion. To validate the generality, we explored the correlation of enrichment score of CAF_{EndMT} and *SPP1*⁺TAM in various cancer types in TCGA by GSVA. Generally, the enrichment score of CAF_{EndMT} was positively associated with that of *SPP1*⁺TAM in absolutely most of cancer types (25 out of 28). Insignificance was only observed in three types of cancer (indicated by red square). We added this analysis result as Supplementary Fig. 8

Correlation of *SPP1*⁺TAM and CAF_{EndMT} estimated in TCGA

10. The relevance of the EMT score to the trajectory of CAFs is unclear given CAFs do not have prominent epithelial feature to begin with. Please clarify.

Response: We thank the reviewer for this great question. We totally agree that CAFs mainly exhibit mesenchymal phenotype, and do not have prominent epithelial feature. However, the EMT is a continuous process from epithelial to mesenchymal state. Furthermore, the dynamic process of EMT is visible when applying scRNA-seq with a series of separating clusters (see below from Carstens *J et al. Cell Rep. 2021*).

Gradually change of EMT score from epithelial to mesenchymal single cell-based clusters

Therefore, although all CAFs are grouped into mesenchymal type, different clusters of CAFs may varied in mesenchymal stages of EMT process, which can be quantified by EMT score. We only

determined the relative EMT score of each CAF to illustrate the potential enrichment of EMT stages in each CAF state. We applied the well-established Hallmark EMT signature (Liberzon A *et al*, *Cell Syst*, 2015) to quantify the relative EMT score of the CAFs cell. To verify the analysis result, we also estimated the EMT score based on another two established EMT signatures (Kalluri R *et al*, *Cell Rep*. 2021; Vasaikar S *et al*, *Br J Cancer*. 2021). Consistently, CAF_{state3} have highest EMT score (see below) compared to those in the other two states.

EMT score of CAFs estimated by another two gene signatures

In addition, when the EMT score of Hallmark EMT signature (Liberzon A *et al*, *Cell Syst*, 2015) in our study were also applied for cancer epithelial, epithelial cells are significantly much lower than other three states of CAFs (all $p < 0.001$) (see below), suggesting the reliability of our analysis.

Comparison of EMT score of CAFs and epithelial cells in our study

Minor comments:

1. In Fig. 4D, it would help to include split channel images to better visualize the overlap between CD163 and α SMA.

Response: We thank the reviewer for this comment. We have revised this part accordingly.

2. The phenotype image in Fig. S53 does not appear to match composite image.

Response: We apologize the confusion. We have revised the figure accordingly (now **Supplementary Fig.7**).

3. 'EndMT' is more often used than 'EndoMT' and the nomenclature used by those who initially reported this phenomenon.

Response: We appreciate this comment and changed all "EndoMT" into "EndMT" through the manuscript.

REVIEWER COMMENTS

Reviewer #1 (Remarks to the Author):

As stated in the first round of review, this is the largest scale and extensive integrated analysis focusing on CAFs and well-suited to demonstrate similarities and differences in CAFs for diverse cancer types.

The study utilized trajectory analysis to suggest origin or plasticity of CAFs. The type of analysis is speculative as the results reflect relative similarities in transcriptome within the data. Analyses inferring interactions or regulons are also speculative, basically representing gene expression abundance in annotated gene sets. While the authors extended marker staining in the revision, they are not convincing evidence for the differentiation or state transition of the CAFs. In the revision, authors stated these limitations in the discussion.

Authors properly addressed issues related to the analysis.

Minor comment: check spelling for the SCENIC package. "SENIC" is used in multiple places.

Reviewer #2 (Remarks to the Author):

The authors have provided some useful revisions to their manuscript, the study is largely the unchanged from the original submission. The main strength of this study remains to be the work done to harmonize scRNAseq data sets across different cancer types and to provide pancancer clusters. This harmonized data will be of interest to the broader community provided it is made available to the community. In re-reviewing the manuscript and the supplementary material is not clear that the harmonized data will be made publicly available.

The main limitation of the analysis is still that the study is largely descriptive. There is no functional validation of any inferences. Only IF images are provided to suggest colocalization of potentially interaction cell types, but these images are limited in number and hard to interpret.

The authors do not provide any analysis to determine if there are conserved pancancer fibroblast-malignant cell interactions. They claim that an analysis of malignant-fibroblast interactions is not

relevant because the malignant cells vary significantly by cancer type. However, given their reported conserved cell types of the fibroblasts (as well as the immune and endothelial cell types), it would be surprising that a ligand-receptor interaction between malignant cells and fibroblasts is not conserved. It is unfortunate that this analysis was not considered.

The authors claim that only one fibroblast cell type (CAF state 3) is prognostic. Because this cell type appears to have a myofibroblast properties, its prognostic significance is not new. Also, claiming that CAF state 3 has a higher EMT score is a confusing use of the EMT signature. The EMT signature relates to epithelial cells acquiring mesenchymal properties. Because CAFs are mesenchymal cells, scoring them along the EMT spectrum is confusing. The authors to emphasize that CAF state 1 and CAF state 2 have a lower EMT score than CAF state 3, but the relevance of things finding is unclear. Are the authors trying to say that CAF state 3 is more like a transformed epithelial cell and if so, why is this important?

Reviewer #3 (Remarks to the Author):

All comments have been addressed satisfactorily. Please consider citing the recent study on functional contributions of CAFs in PDAC by McAndrews et al. (Identification of Functional Heterogeneity of Carcinoma-Associated Fibroblasts with Distinct IL6-Mediated Therapy Resistance in Pancreatic Cancer, Cancer Discovery 2022).

Reviewer #1 (Remarks to the Author):

As stated in the first round of review, this is the largest scale and extensive integrated analysis focusing on CAFs and well-suited to demonstrate similarities and differences in CAFs for diverse cancer types.

The study utilized trajectory analysis to suggest origin or plasticity of CAFs. The type of analysis is speculative as the results reflect relative similarities in transcriptome within the data. Analyses inferring interactions or regulons are also speculative, basically representing gene expression abundance in annotated gene sets. While the authors extended marker staining in the revision, they are not convincing evidence for the differentiation or state transition of the CAFs. In the revision, authors stated these limitations in the discussion.

Authors properly addressed issues related to the analysis.

Response: We greatly appreciate the constructive comments and suggestions to improve our manuscript.

We agreed with the reviewer's comments that the trajectory analysis is based on similarities in transcriptome, while inferring interaction is basically representing gene expression abundance in annotated gene sets. However, Monocle (Qiu X et al, *Nat Methods*, 2017) and CellphoneDB (Efremova M et al, *Nat Protoc*, 2020) are the most widely used software to infer trajectory and cell-cell communications with scRNA-seq data respectively, which has been highly cited, including a series of high-quality papers (e.g., Emont M et al, *Nature*, 2022; Sun Y et al, *Cell*. 2021; Fawcner-Corbett D et al, *Cell*. 2021; Cao J et al, *Nature*. 2019), as well as the recently published scRNA-seq based pan-cancer analysis (e.g., Cheng S et al, *Cell*. 2021). Moreover, some of the inferred interactions and trajectories have been experimental validated through systematical functional investigation (e.g., Centonze A et al, *Nature*, 2020; Garcia-Alonso L et al; *Nat Genet*. 2021), suggesting their reliability and potential value to guide experimental design. In our study, we agree with the reviewer's comments that the experimental analysis is indirect evidence to support the differentiation or state transition of CAFs, which is not strong enough. Therefore, due to the lack of direct experimental evidence in our study, we further discussed the limitation of the software and our study in the revised manuscript (line 550-557).

Minor comment: check spelling for the SCENIC package. "SENIC" is used in multiple places.

Response: We are sorry for this typo and revised it (line 273 and 1075).

Reviewer #2 (Remarks to the Author):

The authors have provided some useful revisions to their manuscript, the study is largely the unchanged from the original submission. The main strengthen of this study remains to be the work done to harmonize scRNAseq data sets across different cancer types and to provide pancancer clusters. This harmonized data will be of interest to the broader community provided it is made available to the community. In re-reviewing the manuscript and the supplementary material is not clear that the harmonized data will be made publicly available.

Response: We appreciate this suggestion, and we have uploaded the integrated matrix file online as GSE210347, description of which has been added in Data and code availability part (line 761-762). Moreover, we have also established an interactive website-based tool (<https://gist-fgl.github.io/sc-caf-atlas/>) which is easy to focus on specific genes and clusters at pan-cancer level. We added this information in the abstract (line 84-85), discussion (line 561-562), and Data and code availability part (line 761-762).

The main limitation of the analysis is still that the study is largely descriptive. There is no functional validation of any inferences. Only IF images are provided to suggest colocalization of potentially interaction cell types, but these images are limited in number and hard to interpret.

Response: We agree that our manuscript is largely descriptive. The main goal of this study is to define the landscape of infiltrating CAFs common to multiple cancer types, similar to recent studies describing the landscape of tumor infiltrating myeloid and T cells (Cheng S *et al. Cell*, 2021; Zheng L *et al. Science*. 2021). We believe that these descriptive studies provide major progress to our understanding of cancer development, provide a global framework to better understand previous mechanistic reports, and will facilitate the design and interpretation of future functional studies. Due to the heterogeneity of CAFs (Sahai E *et al. Nat Rev Cancer*. 2020), high expression of certain specific genes has been noticed as canonical makers of CAF subclusters, and thus used to define these CAF subclusters, like FAP⁺CAF, THY1⁺CAF and ENG⁺CAF. Increasing experimental evidence has revealed the potential bio-functions of these CAFs subclusters in specific diseases or cancers. For example, ENG⁺ (CD105) fibroblasts were permissive for pancreatic tumor growth in vivo (Hutton C *et al. Cancer Cell*. 2021). THY1⁺ (CD90) fibroblasts play important inflammation regulation role in rheumatoid arthritis (Mizoguchi F *et al. Nat Commun*. 2018), FAP⁺THY1⁺ fibroblasts would promote persistent inflammation arthritis (Croft A *et al. Nature*. 2019), FAP⁺ fibroblasts can promote immunosuppression in multiple cancers (liver, lung and pancreatic cancers) (Kraman M *et al. Science*. 2010; McAndrews K *et al. Cancer Discov*. 2022 Yang X *et al. Cancer Res*. 2016) and induced angiogenesis by affecting the balance of pro-angiogenic and anti-angiogenic mediators in lung cancer (Santos A *et al. J Clin Invest*. 2009), and PDGF signaling of CAFs may not only stimulate tumor angiogenesis but also mediate resistance to anti-angiogenic therapy (Crawford Y *et al. Cancer Cell*. 2009; Pietras K *et al. PLoS Med*. 2008). Additionally, the different clusters of CAFs also involved in various function by genes enrichment (**Supplementary Fig. 3e**). Thus, specific marker of CAFs stands for some biofunctions of CAFs in some degree according to current available evidence.

In our study, we illustrated the ubiquitous existence of different CAFs subclusters among multiple cancer types. As the functions of many of these clusters have been revealed in previous mechanistic studies in specific cancer types, we thus suggested the possible shared functions of these clusters in different cancer types. It would improve classification of heterogenous CAFs and facilitate the understanding of them. Since we didn't perform functional validation, we thus modified our expression, cited relevant references of published functional studies, and highlighted the importance of future mechanistic work to avoid substantial overstatement (e.g., line496-497, line 545-557).

Besides, to further validation, we also analyzed scRNAseq data from endothelial cell marker-Cre mouse. Ideally, we can check if there are cells that express both Cre mRNA and fibroblast marker, which could prove the transition from endo to fibroblast (CAF_{EndMT}). Therefore, we used scRNAseq

profile of endothelial cells from previous report (Lukas S. T et al. *Nat Commun*, 2021). In this study, they used the *Cdh5-CreERT2* for endothelial cell lineage tracing from *Cdh5-CreERT2;mT/mG* mice. Thus, Cre recombinase expressing cells (and future cell lineages derived from these cells) have cell membrane-localized EGFP (mG) fluorescence expression. We selected EGFP positive cells from all the dataset, which should be annotated as endothelial cells (see below Fig. a). Then we subcluster EGFP⁺ cells into 6 subclusters. Subclusters 0-4 overall expressed endothelial cell markers, whereas cluster 5 shows that endo markers are faded and mesenchyme-fibroblast related markers (*Acta2*, *Col1a1* and *Col1a2*) are increased (Fig. b and c, below). To identify the EndMT scheme, we divided the cells into *Cdh5*⁺*Acta*⁻(endo), *Cdh5*⁺*Acta*⁺(CAF_{EndMT}), and *Cdh5*⁻*Acta*⁺(mesenchyme). *Cdh5*⁺*Acta*⁻(endo) actively expressed endo markers, but it was faded out at the *Cdh5*⁺*Acta*⁺(mesenchyme). In the *Cdh5*⁻*Acta*⁺(mesenchyme), mesenchyme markers are highly increased compared to *Cdh5*⁺*Acta*⁻(endo). Lastly, *Cdh5*⁺*Acta*⁺(CAF_{EndMT}) expressed both markers endo and mesenchyme (Fig. d, below). Then we did trajectory analysis with subcluster 0-5, because subcluster 6 shows immune cell properties, we removed it in the trajectory analysis (see below Fig. e). In the trajectory cells with EGFP⁺ cells, it clear supported the EndMT (see below Fig. e and f, below). However, it is not tumor-related profile, so we would not illustrate it in the manuscript.

The further analysis of endothelial cells from *Cdh5-CreERT2;mT/mG* mice

Moreover, although we performed mIF to verify the colocalization of different cell type in only 9 patients, which may be limited in number. However, the colocalizations of TAM and CAF_{EndMT} were

observed in all three different cancer types we tested (**Fig. 6e**), and reached statistical significance (**Fig. 6f** and **6g**). In addition, we also conducted spatial transcriptome analysis in 7 patients with colorectal cancer, revealing the significant correlation of CAF_{EndMT} with SPP1⁺TAM with $p < 2 \times 10^{-16}$ (**Fig. 6h**), which verify the colocalization of these two cell types.

The authors do not provide any analysis to determine if there are conserved pancancer fibroblast-malignant cell interactions. They claim that an analysis of malignant-fibroblast interactions is not relevant because the malignant cells vary significantly by cancer type. However, given their reported conserved cell types of the fibroblasts (as well as the immune and endothelial cell types), it would be surprising that a ligand-receptor interaction between malignant cells and fibroblasts is not conserved. It is unfortunate that this analysis was not considered.

Response: We thank this reviewer's suggestion. Since we were concerned about the bias of epithelial cells among different cancer types, we thus separated the epithelial cells into small clusters and identified a subset of ubiquitous clusters shared by different cancer types. On the other hand, given the similarity of CAF among different cancer types, it may be actionable to conduct epithelia-CAF interactions at pan-cancer level. Therefore, we described both epithelia clusters separation (line 223-235) and epithelia-CAF interaction (line 338-347). We first illustrated the similarity/specificity of epithelial clusters among different cancer types (see below and **Supplementary Fig. 2 and Supplementary Table 4 and 5**). A total of 23 epithelial clusters were identified. Unlike TME components, epithelia clusters exhibited bias in terms of both malignant status (e.g., predominant normal in E17 and tumor in E6) and cancer type (e.g., predominant thyroid cancer in E1 and prostate cancer in E6). Similarity among different cancer types was identified in only 5 out of 23 clusters, including E3 (*IRS2*, *KRT6A*), E5 (*CD24*, *STMN1*), E8 (*GKN1*, *MUC5AC*), E9 (*PHGR1*, *TFF3*), E10 (*TFF3*, *TPO*) and E13 (*VTN* and *ITIH5*). Interestingly, *IRS2* and *CD24* were widely present in epithelia and exhibited the strongest expression in E3 and E5 cluster respectively, suggesting the ubiquitous activated insulin signaling and "don't eat me" signal (*Barkal A et al. Nature. 2019*) shared by different cancer types.

Subsequently, epithelia-CAF interactions were estimated through CellphoneDB. Interestingly epithelia have the most prolific interactions with fibroblasts than with other TME components. Therefore, we estimated the communications between three CAF states and each epithelia cluster. Not surprisingly, CAF_{state3} also exhibited more interactions with epithelia than CAF_{state1/2}, especially with E3/E18 (clusters shared by all cancer types), E4 (dominant in digestive system tumor), and E18 (dominant in breast and ovarian cancer) (see below and **Supplementary 5d**). Moreover, we focused on the crosstalk between each CAF state with the epithelia clusters that exhibited similarities across cancer types (i.e., E3, E5, E8, E9, E19, and E13) (see below and **Supplementary Fig. 5e**). A series of ligand-receptor pairs were identified, which are involved in cancer related pathways, including EGFR (e.g., EGFR_TGFB1), NOTCH (e.g., JAG1_NOTCH2/3), WNT pathways (e.g., FZD6_WNT5A) (see below and **Supplementary Fig. 5f**). We added these data in the main text of

the revised manuscript (line 338-347).

Interactions profile between CAF states and epithelial clusters

The authors claim that only one fibroblast cell type (CAF state 3) is prognostic. Because this cell type appears to have a myofibroblast properties, its prognostic significance is not new. Also, claiming that CAF state 3 has a higher EMT score is a confusing use of the EMT signature. The EMT signature relates to epithelial cells acquiring mesenchymal properties. Because CAFs are mesenchymal cells, scoring them along the EMT spectrum is confusing. The authors to emphasize that CAF state 1 and CAF state 2 have a lower EMT score than CAF state 3, but the relevance of things finding is unclear. Are the authors trying to say that CAF state 3 is more like a transformed epithelial cell and if so, why is this important?

Response: We thank the reviewer for this comment. First, we completely agree with the reviewer that prognostic significance of myofibroblast properties is not new. However, according to our analysis, CAF_{state3} is dominated by inflammatory CAF (CAF_{infla}) and adipogenic CAF (CAF_{adi}), whereas CAF_{state2} is dominated by myofibroblast (CAF_{myo}) (Fig. 2f and line 284-285). Although the prognostic value of CAF_{myo} in non-immunotherapy treatment has been described, here we identified a novel prognostic value of CAF_{state3}, rather than CAF_{state2}, in three independent immunotherapy cohorts (Fig. 3g and Supplementary Fig. 6). Moreover, the expression of EMT markers in CAFs does not necessarily implies that CAFs have an epithelial origin or epithelial transformed. For instance, classic EMT transcription factors (ZEB1, TWIST, SNAIL) are expressed at higher levels in CAFs compared to tissue fibroblasts, which was associated with progression, metastasis, and drug-resistance (Rong Fu *et al*, *Nat Commun.* 2019; Josep Baulida, *Mol Oncol.* 2017).

Second, we agree with the reviewer that CAFs are mesenchymal cells, and do not have prominent epithelial features. However, EMT is a continuous process from epithelial to mesenchymal state. Furthermore, the dynamic process of EMT is visible when applying scRNA-seq with a series of separating clusters (see below from Carstens J *et al*. *Cell Rep.* 2021).

Gradually change of EMT score from epithelial to mesenchymal single cell-based clusters

Therefore, although all CAFs are grouped into mesenchymal type, different clusters of CAFs may varied in mesenchymal stages of EMT process, which can be quantified by EMT score. We only determined the relative EMT score of each CAF to illustrate the potential enrichment of EMT stages in each CAF state. We applied the well-established Hallmark EMT signature (Liberzon A *et al*, *Cell Syst*, 2015) to quantify the relative EMT score of the CAFs cell. To verify the analysis result, we also estimated the EMT score based on another two established EMT signatures (Kalluri R *et al*, *Cell Rep*. 2021; Vasaikar S *et al*, *Br J Cancer*. 2021). Consistently, CAF_{state3} have highest EMT score (see below) compared to those in the other two states.

EMT score of CAFs estimated by another two gene signatures

In addition, when the EMT score of Hallmark EMT signature (Liberzon A *et al*, *Cell Syst*, 2015) in our study were also applied for cancer epithelial, epithelial cells are significantly lower than other three states of CAFs (all $p < 0.001$) (see below), suggesting the reliability of our analysis.

Comparison of EMT score of CAFs and epithelial cells in our study

Reviewer #3 (Remarks to the Author):

All comments have been addressed satisfactorily. Please consider citing the recent study on functional contributions of CAFs in PDAC by McAndrews et al. (Identification of Functional Heterogeneity of Carcinoma-Associated Fibroblasts with Distinct IL6-Mediated Therapy Resistance in Pancreatic Cancer, Cancer Discovery 2022).

Response: We greatly appreciate the constructive comments and suggestions to improve our manuscript. And we have added this citation to support our finding on FAP⁺ CAF (line 514)

REVIEWERS' COMMENTS

Reviewer #1 (Remarks to the Author):

My comments have been addressed in the previous revision. Authors acknowledged limitations of the study, i.e. mostly bioinformatic inferences without functional validation_ in the discussion.

Reviewer #2 (Remarks to the Author):

Authors have addressed some my comments but their work still has little evidence of rigor or reproducibility for key findings. However, the authors have appropriately toned down the significance of the findings.

Reviewer #1 (Remarks to the Author):

As stated in the first round of review, this is the largest scale and extensive integrated analysis focusing on CAFs and well-suited to demonstrate similarities and differences in CAFs for diverse cancer types.

The study utilized trajectory analysis to suggest origin or plasticity of CAFs. The type of analysis is speculative as the results reflect relative similarities in transcriptome within the data. Analyses inferring interactions or regulons are also speculative, basically representing gene expression abundance in annotated gene sets. While the authors extended marker staining in the revision, they are not convincing evidence for the differentiation or state transition of the CAFs. In the revision, authors stated these limitations in the discussion.

Authors properly addressed issues related to the analysis.

Response: We greatly appreciate the constructive comments and suggestions to improve our manuscript.

We agreed with the reviewer's comments that the trajectory analysis is based on similarities in transcriptome, while inferring interaction is basically representing gene expression abundance in annotated gene sets. However, Monocle (Qiu X et al, *Nat Methods*, 2017) and CellphoneDB (Efremova M et al, *Nat Protoc*, 2020) are the most widely used software to infer trajectory and cell-cell communications with scRNA-seq data respectively, which has been highly cited, including a series of high-quality papers (e.g., Emont M et al, *Nature*, 2022; Sun Y et al, *Cell*. 2021; Fawcner-Corbett D et al, *Cell*. 2021; Cao J et al, *Nature*. 2019), as well as the recently published scRNA-seq based pan-cancer analysis (e.g., Cheng S et al, *Cell*. 2021). Moreover, some of the inferred interactions and trajectories have been experimental validated through systematical functional investigation (e.g., Centonze A et al, *Nature*, 2020; Garcia-Alonso L et al; *Nat Genet*. 2021), suggesting their reliability and potential value to guide experimental design. In our study, we agree with the reviewer's comments that the experimental analysis is indirect evidence to support the differentiation or state transition of CAFs, which is not strong enough. Therefore, due to the lack of direct experimental evidence in our study, we further discussed the limitation of the software and our study in the revised manuscript (line 550-557).

Minor comment: check spelling for the SCENIC package. "SENIC" is used in multiple places.

Response: We are sorry for this typo and revised it (line 273 and 1075).

Reviewer #2 (Remarks to the Author):

The authors have provided some useful revisions to their manuscript, the study is largely the unchanged from the original submission. The main strengthen of this study remains to be the work done to harmonize scRNAseq data sets across different cancer types and to provide pancancer clusters. This harmonized data will be of interest to the broader community provided it is made available to the community. In re-reviewing the manuscript and the supplementary material is not clear that the harmonized data will be made publicly available.

Response: We appreciate this suggestion, and we have uploaded the integrated matrix file online as GSE210347, description of which has been added in Data and code availability part (line 761-762). Moreover, we have also established an interactive website-based tool (<https://gist-fgl.github.io/sc-caf-atlas/>) which is easy to focus on specific genes and clusters at pan-cancer level. We added this information in the abstract (line 84-85), discussion (line 561-562), and Data and code availability part (line 761-762).

The main limitation of the analysis is still that the study is largely descriptive. There is no functional validation of any inferences. Only IF images are provided to suggest colocalization of potentially interaction cell types, but these images are limited in number and hard to interpret.

Response: We agree that our manuscript is largely descriptive. The main goal of this study is to define the landscape of infiltrating CAFs common to multiple cancer types, similar to recent studies describing the landscape of tumor infiltrating myeloid and T cells (Cheng S *et al. Cell*, 2021; Zheng L *et al. Science*. 2021). We believe that these descriptive studies provide major progress to our understanding of cancer development, provide a global framework to better understand previous mechanistic reports, and will facilitate the design and interpretation of future functional studies. Due to the heterogeneity of CAFs (Sahai E *et al. Nat Rev Cancer*. 2020), high expression of certain specific genes has been noticed as canonical makers of CAF subclusters, and thus used to define these CAF subclusters, like FAP⁺CAF, THY1⁺CAF and ENG⁺CAF. Increasing experimental evidence has revealed the potential bio-functions of these CAFs subclusters in specific diseases or cancers. For example, ENG⁺ (CD105) fibroblasts were permissive for pancreatic tumor growth in vivo (Hutton C *et al. Cancer Cell*. 2021). THY1⁺ (CD90) fibroblasts play important inflammation regulation role in rheumatoid arthritis (Mizoguchi F *et al. Nat Commun*. 2018), FAP⁺THY1⁺ fibroblasts would promote persistent inflammation arthritis (Croft A *et al. Nature*. 2019), FAP⁺ fibroblasts can promote immunosuppression in multiple cancers (liver, lung and pancreatic cancers) (Kraman M *et al. Science*. 2010; McAndrews K *et al. Cancer Discov*. 2022 Yang X *et al. Cancer Res*. 2016) and induced angiogenesis by affecting the balance of pro-angiogenic and anti-angiogenic mediators in lung cancer (Santos A *et al. J Clin Invest*. 2009), and PDGF signaling of CAFs may not only stimulate tumor angiogenesis but also mediate resistance to anti-angiogenic therapy (Crawford Y *et al. Cancer Cell*. 2009; Pietras K *et al. PLoS Med*. 2008). Additionally, the different clusters of CAFs also involved in various function by genes enrichment (**Supplementary Fig. 3e**). Thus, specific marker of CAFs stands for some biofunctions of CAFs in some degree according to current available evidence.

In our study, we illustrated the ubiquitous existence of different CAFs subclusters among multiple cancer types. As the functions of many of these clusters have been revealed in previous mechanistic studies in specific cancer types, we thus suggested the possible shared functions of these clusters in different cancer types. It would improve classification of heterogenous CAFs and facilitate the understanding of them. Since we didn't perform functional validation, we thus modified our expression, cited relevant references of published functional studies, and highlighted the importance of future mechanistic work to avoid substantial overstatement (e.g., line496-497, line 545-557).

Besides, to further validation, we also analyzed scRNAseq data from endothelial cell marker-Cre mouse. Ideally, we can check if there are cells that express both Cre mRNA and fibroblast marker, which could prove the transition from endo to fibroblast (CAF_{EndMT}). Therefore, we used scRNAseq

profile of endothelial cells from previous report (Lukas S. T et al. *Nat Commun*, 2021). In this study, they used the *Cdh5-CreERT2* for endothelial cell lineage tracing from *Cdh5-CreERT2;mT/mG* mice. Thus, Cre recombinase expressing cells (and future cell lineages derived from these cells) have cell membrane-localized EGFP (mG) fluorescence expression. We selected EGFP positive cells from all the dataset, which should be annotated as endothelial cells (see below Fig. a). Then we subcluster EGFP⁺ cells into 6 subclusters. Subclusters 0-4 overall expressed endothelial cell markers, whereas cluster 5 shows that endo markers are faded and mesenchyme-fibroblast related markers (*Acta2*, *Col1a1* and *Col1a2*) are increased (Fig. b and c, below). To identify the EndMT scheme, we divided the cells into *Cdh5*⁺*Acta*⁻(endo), *Cdh5*⁺*Acta*⁺(CAF_{EndMT}), and *Cdh5*⁻*Acta*⁺(mesenchyme). *Cdh5*⁺*Acta*⁻(endo) actively expressed endo markers, but it was faded out at the *Cdh5*⁺*Acta*⁺(mesenchyme). In the *Cdh5*⁻*Acta*⁺(mesenchyme), mesenchyme markers are highly increased compared to *Cdh5*⁺*Acta*⁻(endo). Lastly, *Cdh5*⁺*Acta*⁺(CAF_{EndMT}) expressed both markers endo and mesenchyme (Fig. d, below). Then we did trajectory analysis with subcluster 0-5, because subcluster 6 shows immune cell properties, we removed it in the trajectory analysis (see below Fig. e). In the trajectory cells with EGFP⁺ cells, it clear supported the EndMT (see below Fig. e and f, below). However, it is not tumor-related profile, so we would not illustrate it in the manuscript.

The further analysis of endothelial cells from *Cdh5-CreERT2;mT/mG* mice

Moreover, although we performed mIF to verify the colocalization of different cell type in only 9 patients, which may be limited in number. However, the colocalizations of TAM and CAF_{EndMT} were

observed in all three different cancer types we tested (**Fig. 6e**), and reached statistical significance (**Fig. 6f** and **6g**). In addition, we also conducted spatial transcriptome analysis in 7 patients with colorectal cancer, revealing the significant correlation of CAF_{EndMT} with SPP1⁺TAM with $p < 2 \times 10^{-16}$ (**Fig. 6h**), which verify the colocalization of these two cell types.

The authors do not provide any analysis to determine if there are conserved pancancer fibroblast-malignant cell interactions. They claim that an analysis of malignant-fibroblast interactions is not relevant because the malignant cells vary significantly by cancer type. However, given their reported conserved cell types of the fibroblasts (as well as the immune and endothelial cell types), it would be surprising that a ligand-receptor interaction between malignant cells and fibroblasts is not conserved. It is unfortunate that this analysis was not considered.

Response: We thank this reviewer's suggestion. Since we were concerned about the bias of epithelial cells among different cancer types, we thus separated the epithelial cells into small clusters and identified a subset of ubiquitous clusters shared by different cancer types. On the other hand, given the similarity of CAF among different cancer types, it may be actionable to conduct epithelia-CAF interactions at pan-cancer level. Therefore, we described both epithelia clusters separation (line 223-235) and epithelia-CAF interaction (line 338-347). We first illustrated the similarity/specificity of epithelial clusters among different cancer types (see below and **Supplementary Fig. 2 and Supplementary Table 4 and 5**). A total of 23 epithelial clusters were identified. Unlike TME components, epithelia clusters exhibited bias in terms of both malignant status (e.g., predominant normal in E17 and tumor in E6) and cancer type (e.g., predominant thyroid cancer in E1 and prostate cancer in E6). Similarity among different cancer types was identified in only 6 out of 23 clusters, including E3 (*IRS2*, *KRT6A*), E5 (*CD24*, *STMN1*), E8 (*GKN1*, *MUC5AC*), E9 (*PHGR1*, *TFF3*), E10 (*TFF3*, *TPO*) and E13 (*VTN* and *ITIH5*). Interestingly, *IRS2* and *CD24* were widely present in epithelia and exhibited the strongest expression in E3 and E5 cluster respectively, suggesting the ubiquitous activated insulin signaling and "don't eat me" signal (*Barkal A et al. Nature. 2019*) shared by different cancer types.

Subsequently, epithelia-CAF interactions were estimated through CellphoneDB. Interestingly epithelia have the most prolific interactions with fibroblasts than with other TME components. Therefore, we estimated the communications between three CAF states and each epithelia cluster. Not surprisingly, CAF_{state3} also exhibited more interactions with epithelia than CAF_{state1/2}, especially with E3/E18 (clusters shared by all cancer types), E4 (dominant in digestive system tumor), and E18 (dominant in breast and ovarian cancer) (see below and **Supplementary 5d**). Moreover, we focused on the crosstalk between each CAF state with the epithelia clusters that exhibited similarities across cancer types (i.e., E3, E5, E8, E9, E19, and E13) (see below and **Supplementary Fig. 5e**). A series of ligand-receptor pairs were identified, which are involved in cancer related pathways, including EGFR (e.g., EGFR_TGFB1), NOTCH (e.g., JAG1_NOTCH2/3), WNT pathways (e.g., FZD6_WNT5A) (see below and **Supplementary Fig. 5f**). We added these data in the main text of

the revised manuscript (line 338-347).

Interactions profile between CAF states and epithelial clusters

The authors claim that only one fibroblast cell type (CAF state 3) is prognostic. Because this cell type appears to have a myofibroblast properties, its prognostic significance is not new. Also, claiming that CAF state 3 has a higher EMT score is a confusing use of the EMT signature. The EMT signature relates to epithelial cells acquiring mesenchymal properties. Because CAFs are mesenchymal cells, scoring them along the EMT spectrum is confusing. The authors to emphasize that CAF state 1 and CAF state 2 have a lower EMT score than CAF state 3, but the relevance of things finding is unclear. Are the authors trying to say that CAF state 3 is more like a transformed epithelial cell and if so, why is this important?

Response: We thank the reviewer for this comment. First, we completely agree with the reviewer that prognostic significance of myofibroblast properties is not new. However, according to our analysis, CAF_{state3} is dominated by inflammatory CAF (CAF_{infla}) and adipogenic CAF (CAF_{adi}), whereas CAF_{state2} is dominated by myofibroblast (CAF_{myo}) (Fig. 2f and line 284-285). Although the prognostic value of CAF_{myo} in non-immunotherapy treatment has been described, here we identified a novel prognostic value of CAF_{state3}, rather than CAF_{state2}, in three independent immunotherapy cohorts (Fig. 3g and Supplementary Fig. 6). Moreover, the expression of EMT markers in CAFs does not necessarily implies that CAFs have an epithelial origin or epithelial transformed. For instance, classic EMT transcription factors (ZEB1, TWIST, SNAIL) are expressed at higher levels in CAFs compared to tissue fibroblasts, which was associated with progression, metastasis, and drug-resistance (Rong Fu *et al*, *Nat Commun.* 2019; Josep Baulida, *Mol Oncol.* 2017).

Second, we agree with the reviewer that CAFs are mesenchymal cells, and do not have prominent epithelial features. However, EMT is a continuous process from epithelial to mesenchymal state. Furthermore, the dynamic process of EMT is visible when applying scRNA-seq with a series of separating clusters (see below from Carstens J *et al.* *Cell Rep.* 2021).

Gradually change of EMT score from epithelial to mesenchymal single cell-based clusters

Therefore, although all CAFs are grouped into mesenchymal type, different clusters of CAFs may varied in mesenchymal stages of EMT process, which can be quantified by EMT score. We only determined the relative EMT score of each CAF to illustrate the potential enrichment of EMT stages in each CAF state. We applied the well-established Hallmark EMT signature (Liberzon A *et al*, *Cell Syst*, 2015) to quantify the relative EMT score of the CAFs cell. To verify the analysis result, we also estimated the EMT score based on another two established EMT signatures (Kalluri R *et al*, *Cell Rep*. 2021; Vasaikar S *et al*, *Br J Cancer*. 2021). Consistently, CAF_{state3} have highest EMT score (see below) compared to those in the other two states.

EMT score of CAFs estimated by another two gene signatures

In addition, when the EMT score of Hallmark EMT signature (Liberzon A *et al*, *Cell Syst*, 2015) in our study were also applied for cancer epithelial, epithelial cells are significantly lower than other three states of CAFs (all $p < 0.001$) (see below), suggesting the reliability of our analysis.

Comparison of EMT score of CAFs and epithelial cells in our study

Reviewer #3 (Remarks to the Author):

All comments have been addressed satisfactorily. Please consider citing the recent study on functional contributions of CAFs in PDAC by McAndrews et al. (Identification of Functional Heterogeneity of Carcinoma-Associated Fibroblasts with Distinct IL6-Mediated Therapy Resistance in Pancreatic Cancer, Cancer Discovery 2022).

Response: We greatly appreciate the constructive comments and suggestions to improve our manuscript. And we have added this citation to support our finding on FAP⁺ CAF (line 514)